# Creating the funerary landscape of Eastern Sudan

**Stefano Costanzo** [1]*, **Filippo Brandolini**[2]*, **Habab Idriss Ahmed**[3], **Andrea Zerboni** [4]*, **Andrea Manzo**[1]

**1** Dipartimento Asia, Africa e Mediterraneo, Università degli Studi di Napoli "L'Orientale", Napoli, Italy, **2** McCord Centre for Landscape - School of History, Classics and Archaeology, Newcastle University, Newcastle upon Tyne, United Kingdom, **3** NCAM—National Corporation for Antiquities and Museums, Khartoum, Sudan, **4** Dipartimento di Scienze della Terra 'A. Desio', Università degli Studi di Milano, Milano, Italy

* ste.costanzo92@gmail.com (SC); filippo.brandolini@newcastle.ac.uk (FB); andrea.zerboni@unimi.it (AZ)

## Abstract

Funerary landscapes are eminent results of the relationship between environments and superstructural human behavior, spanning over wide territories and growing over centuries. The comprehension of such cultural palimpsests needs substantial research efforts in the field of human ecology. The funerary landscape of the semi-arid region of Kassala (Eastern Sudan) represents a solid example. Therein, geoarchaeological surveys and the creation of a desk-based dataset of thousands of diachronic funerary monuments (from early tumuli up to modern Beja people islamic tombs) were achieved by means of fieldwork and remote sensing over an area of $\sim 4100$ km$^2$. The wealth of generated information was employed to decipher the spatial arrangement of sites and monuments using Point Pattern Analysis. The enormous number of monuments and their spatial distribution are here successfully explained using, for the first time in archaeology, the Neyman-Scott Cluster Process, hitherto designed for cosmology. Our study highlights the existence of a built funerary landscape with galaxy-like aggregations of monuments driven by multiple layers of societal behavior. We suggest that the distribution of monuments was controlled by a synthesis of opportunistic geological constraints and cultural superstructure, conditioned by the social memory of the Beja people who have inhabited the region for two thousand years and still cherish the ancient tombs as their own kin's.

## 1 Introduction

Among the many facets of archaeological vestiges, burials and funerary monuments are common. In some cases, they are not simply inhumations, but rather testify complex societal behaviours; funerary practices thus symbolize a specific type of human interaction with the landscape [1]. In the last millennia, human groups created complex funerary landscapes that sometimes are the palimpsest of funerary monuments belonging to different cultural phases of prehistory and history [2]. Such built environments are especially well preserved and evident in arid lands because of the lack of vegetation cover. During the past decades, several arid-land archaeological projects pursued dedicated studies on the topic, each presenting case-specific

**Data Availability Statement:** This new DOI represents all accessible versions of our R script code, and will always resolve to the latest one. The latest and definitive version, ready for the upload, was validated during the last stage of the peer

reviewing process. It already has a reserved DOI (https://doi.org/10.5281/zenodo.4384806) and will contain the references to the main PLOS publication R script Code and dataset: https://doi.org/10.5281/zenodo.4384806.

**Funding:** This research was made possible through the support of the fundings awarded to the Italian Archaeological Expedition to the Eastern Sudan (IAEES) by the University of Naples "L'Orientale", the ISMEO — Associazione Internazionale di Studi sul Mediterraneo e l'Oriente, and the Italian Ministry of Foreign Affairs and International Cooperation. The IAEES is also supported by the Regional Government of the Kassala State, Sudan. Additional financial support was provided by Italian Ministry of Education, University, and Research through the project 'Dipartimenti di Eccellenza 2018-2022' (WP4 — Risorse del Patrimonio Culturale), awarded to the Dipartimento di Scienze della Terra 'A. Desio' (University of Milan, Italy). The funders had no role in study design, data collection and analysis, decision to publish, or preparation of the manuscript.

**Competing interests:** The authors have declared that no competing interests exist.

approaches that focused on GIS-based spatial analysis of large areas [3–5] or geoarchaeological characterizations of relatively small sites [6–9]. In this study, we focus on Sudan, whose territory is archaeologically prosperous with countless funerary monumental manifestations of the historical civilizations of the Nile Valley, which has traditionally been the most investigated region by explorers and scholars [10]. The latter often described sites as palimpsests of evidence dating to different epochs up to today, as in the case of Al Khiday (south of the Khartoum-Omdurman conurbation) [11–13] or the Abu Hamed district (4th cataract of the Nile) [14,15], where countless and vastly diachronic burials are stratified at the very same localities. On the contrary, regions far from the Nile Valley remained, until recently, relatively unexplored due to generally lesser academic interest and lack of infrastructure [10,16]. Nonetheless, they are prone to destruction caused by conflicts and land mismanagement like many other more famed North-African sites [17–19].

One example is in the Eastern Sudan, specifically the hilly Kassala region and Eritrean borderland, where thousands of raised funerary monuments are an integral part of the landscape (Fig 1). Scanty information is available about ancient funerary practice and distribution of monuments in the region, but it is known that they can be ascribed to different periods and cultures [10]. Some of them are known as tumuli: relatively simple raised structures, widespread throughout African prehistory and history [20–22], classified into several variants according to the characteristics of their raised portion. Those in Eastern Sudan are of uncertain origin and have been chronologically attributed to the 1st mill. CE [10]. Another category of funerary monuments is related to medieval Islam [23–25]. These visually striking and generally well-preserved tombs are known as *qubbas*, a term that in the pan-Arab world refers to Islamic tombs and shrines. Although the *qubbas* in the region are extremely numerous, their origins and architectural style have been a topic of merely sporadic debate. Initial investigations attributed them to ancient groups of Beja people peacefully converted to Islam [23] or dispersed descendants of the wealthy Arab inhabitants of the ruined town of Badi (Eiri Island) [24]. More recently, some interpretations ascribed their appearance to the resolution of centuries of conflict between Arab miners from Upper Egypt and local Beja people, which led to the islamization of the latter [25].

As it is increasingly being highlighted by many scholars working in the arid belt of North Africa, Near East and Arabia, the interpretation of funerary landscapes and monumental contexts has major implications. Desert are recurrently addressed as empty spaces, but therein archaeological stone monuments represent a major constituent of the cultural landscapes and a glaring mark of the Anthropocene. Such complex archaeological landscapes also represent proxy data for past different environmental settings [28–32]. In this regard, the objective of our study is the comprehensive interpretation of the environmental and cultural factors underlying the genesis and evolution of the funerary landscape of Eastern Sudan. This was achieved by merging geoarchaeological and GIS analyses with advanced spatial statistics into an easily replicable step-by-step workflow. In our analysis we do not examine the tumuli and *qubbas* as discrete *burialscapes* [1], but rather focus on their relationship with the region's physical and cultural geography as a whole, where they blended with the staggering natural setting to create a monumental manifestation of human agency within the landscape. We determined and quantified the environmental and societal controls on their distribution and density, offering an interpretation of the diachronic cultural stratification of the funerary landscape.

## 2 Geography and archaeology in the region of Kassala

The Kassala province is an administrative region of the semi-arid far eastern Sudanese Sahel, extending from the western Butana floodplains to the mountainous Eritrean borderland. For

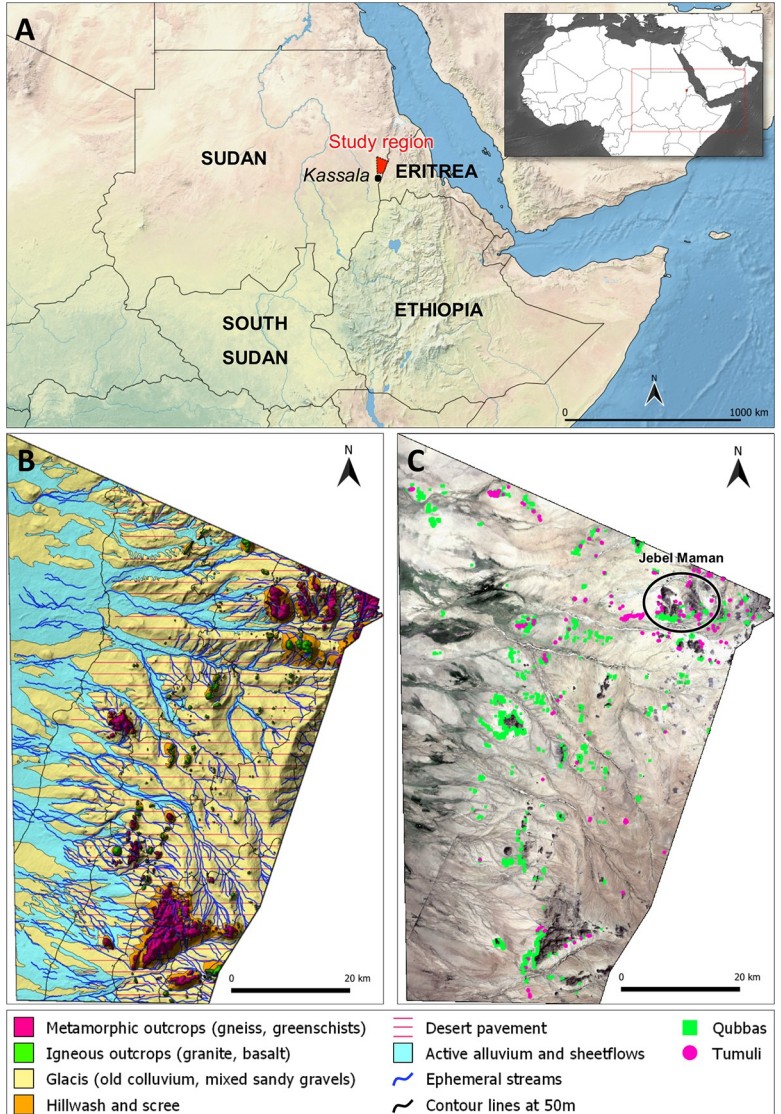

**Fig 1. Setting of the study region and dataset.** (A) Overview of the Study Region and its supra-regional setting. Open-Access image courtesy of Natural Earth [26]. (B) Geomorphology of the study area, modified after [27]. (C) Satellite image of the study area with dataset superimposition. Open-Access Landsat-8 image courtesy of the U.S. Geological Survey.

this research we selected an area of ~4100 km², encircled by the Sudanese-Eritrean border to the east and by arbitrary limits to the north and west/south-west, where the southern edges of the Red Sea Hills and the Gash river's endorheic fan depression are located (Fig 1).

The landscape features a vast and gentle pediment connecting the Gash River's eastern bank to the foothills of the western Eritrean Highlands. The pediment is dotted with twisted bare rock outcrops of Precambrian gneiss basement and Neoproterozoic igneous intrusions and greenschist assemblages, all related to the orogeny of the Arabian-Nubian Shield [33,34]. The rocky outcrops were heavily fractured and cut into inselbergs by regional tectonics and extensive weathering [27]. Several ephemeral streams originate from the hillslopes and converge into large *wadis* (dry, seasonally-active riverbeds) after carving into the hillwash and the

Quaternary deposits, following intricate dendritic patterns and coalescing downslope towards the Gash River [27]. The erosive power of the *wadis* shaped an alternation of shallow valleys and ridges [27]. In general, the extant landscape is the result of deep rock weathering, fluvial/colluvial phenomena and aeolian dynamics triggered by humid/arid climatic shifts that acted around minor regional tectonic adjustments during the Quaternary [27,35].

Archaeological records from the Gash river's floodplains and the Atbara Valley suggest that Eastern Sudanese human occupation may date back to the Palaeolithic, with a flourishing of cultures happening particularly during the Holocene [10,36–44]. Hundreds of prehistoric, proto-historic and later archaeological sites are located along the Gash River's palaeochannels and at the foothills of isolated rocky outcrops ([10] and references cited therein). During the Holocene, people living in the region have gradually shifted from a sedentary lifestyle, favoured by former humid climate that allowed rainfed cultivation [45], to a semi-nomadic pastoral economy following the progressive aridification of the region. Nowadays, the Beja people are the most wide-spread ethnic group of the region, distributed within the Egyptian Eastern Desert, the Kassala region of Sudan and the Red Sea Hills and Eritrean territories. Ancient written sources and social memory suggest that they settled the region well before 2000 years ago [46]. Despite the region's modern agricultural economy, Beja people still preserve a semi-nomadic lifestyle.

The historically sparse presence of people within such a vast land left scanty evidence of settlements yet abundant occurrence of burials clustered around topographic landmarks. Indeed, it has long been known that the hillslopes and hillocks of the Eastern Desert and Red Sea Hills are dotted with raised funerary monuments [10,23,24,47,48]. As previously mentioned, such monuments belong to two main categories: *tumuli* (mounds) and *qubbas*. *Tumuli* (Fig 2) can be earthen- or stone-built, very small (ca. 0.5 m in diameter) or exceeding 20 m in diameter. The raised structures occur as rings, disks or rounded cones. They are usually found in small groups or isolated, although sometimes they can be found in large clusters (Fig 3). *Qubbas* in Eastern Sudan are square structures, usually measuring up to 5 m per side and 2 m in height, built with superimposed unplastered flat slabs of foliated metamorphic rocks (Fig 2) and clustered into groups of up to thousands of elements at the foothills of isolated gneiss/schist hills or on top of small hillocks and ridges (Fig 3). According to previous observations [10,23–25,47], they occur in several stylistic variations, which appear to be related to geographical and chronological differences: simple, unadorned two-storey domed square buildings (16th-17th century AD, type 1); more refined structures adorned with mud and plaster (16th-17th century AD, type 2); much later and quite different shrines occurring as large single tombs of 'holy men' (up to 20th century AD, type 3) [25]. Their spatial extent was thought to be comprised between the Jebel Maman hills (16.28N, 36.8E)(Fig 1), the settlement of Tohamiyam (18.34N, 36.54E) and the Gulf of Agig, along the shorelines of the Red Sea (18.17N, 38.27E) [23–25], yet we located the major concentration of burials around the previously unsurveyed hills stretching 60 km south of Jebel Maman (Fig 1).

## 3 Material and methods

### 3.1 Dataset

The monuments' dataset has been compiled using a remote sensing approach. Open access satellite imagery was manually explored through the plugin QuickMapServices [49] in the GIS software QGIS 3.4 [50]. A 5x5 km grid was superimposed to the study area (ROI) to guarantee an orderly examination, gradually marking the completed squares. Adequate symbols were used for different kinds of archaeological evidence, manually and individually pinpointing all the recognizable features. Terrain surveys were conducted in sampled areas of the ROI to validate the remotely detected archaeological features. The desk-created dataset comprises 783

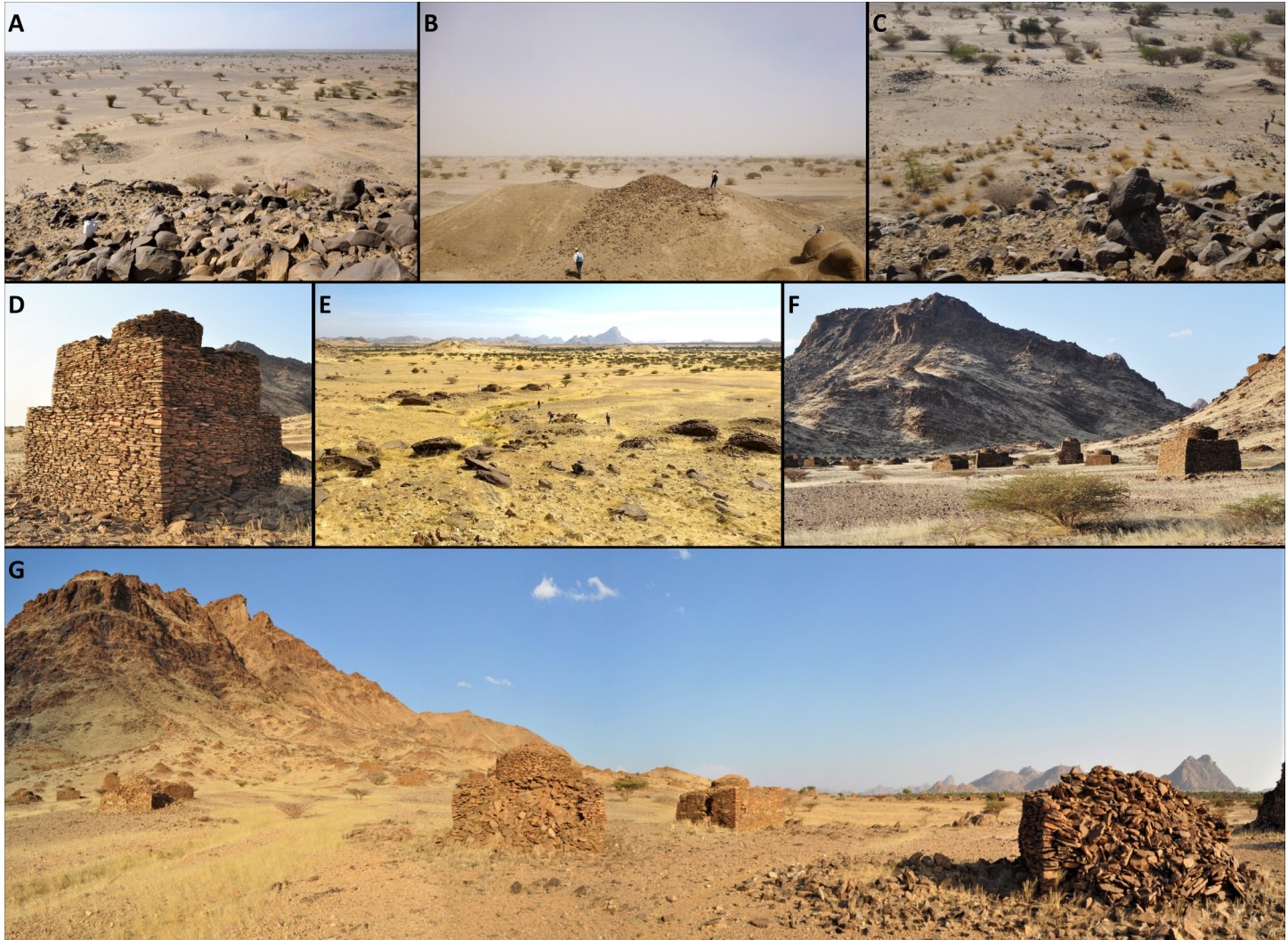

**Fig 2. Field pictures of the funerary monuments.** (A–C) Examples of foothill *tumuli*. (D) A well preserved two-storey *qubba*. (E–G) Landscape views of scatters of *qubbas* around the Jebel Maman.

tumuli and 10274 *qubbas* over an area of ∼4100 km². The tumuli include similar amounts of conical heaps and flat rings varieties. They are mostly found as either small, tight foothill clusters or as large open-plain aggregations that preserve larger inter-monument space. The *qubbas* only occur as type 1 (see 1.1) and always form tight clusters, with rare exceptions.

No permits were required for the described study, which complied with all relevant regulations. The field surveys were carried out under the supervision of NCAM–National Corporation for Antiquities and Museums (Khartoum, Sudan) officers. No archaeological excavations nor collections of archaeological material were performed. The desk-compiled, fully georeferenced archaeological dataset is publicly available at https://doi.org/10.5281/zenodo.4576163.

## 3.2 Point pattern analysis

Point pattern analysis (PPA) is the study of the spatial arrangements of points in space. The application of PPA in landscape archaeology is growing popular for investigating both

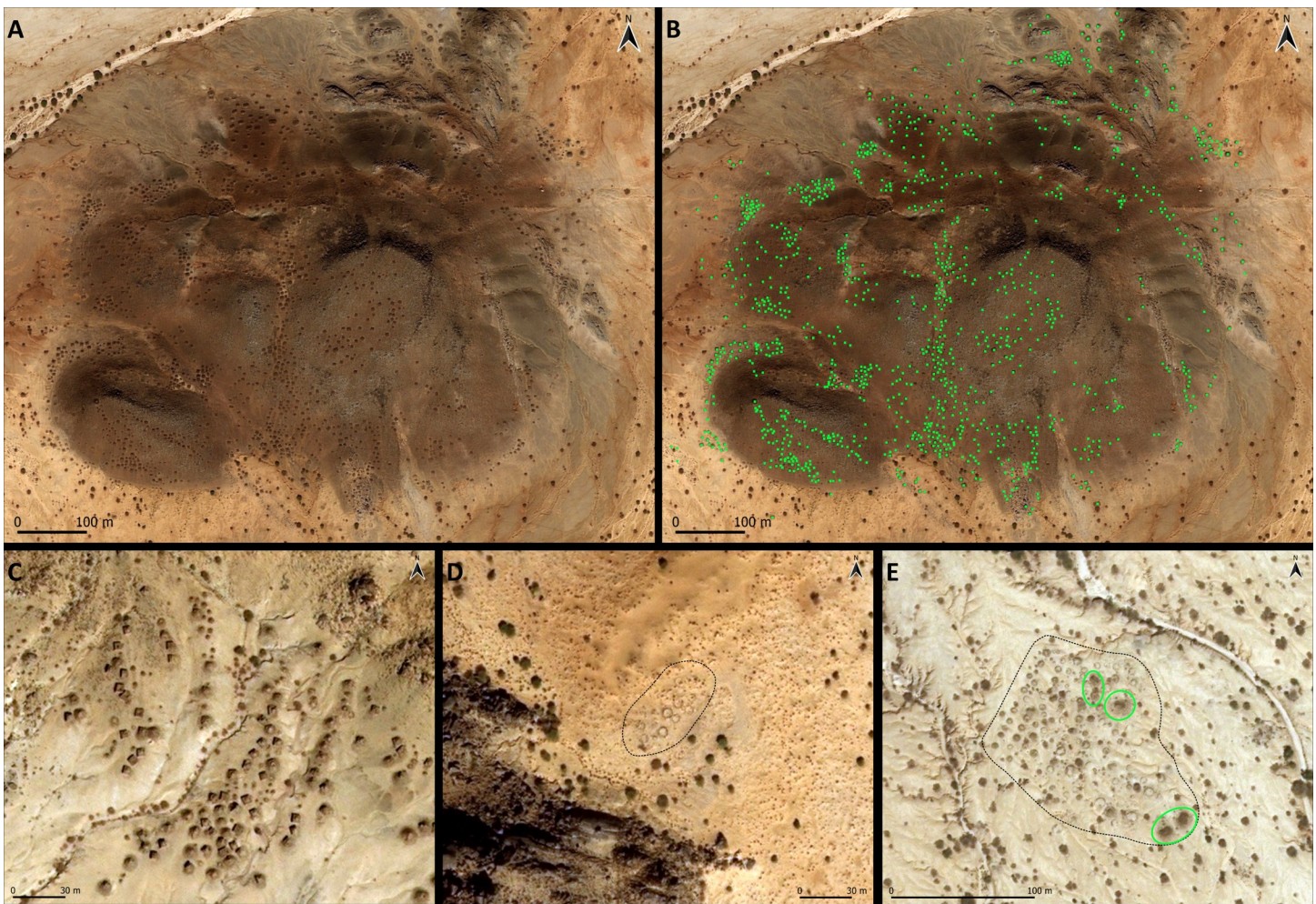

**Fig 3. Satellite imagery illustrating the distribution of funerary monuments.** (A, B) Aggregation of 1195 *qubbas* around and atop a small rocky outcrop. (C) Well preserved clusters of *qubbas* at the foothill of Jebel Maman. (D) A small foothill cluster of ring-shaped tumuli. (E) A large foothill cluster of heap- and ring-type tumuli; a few *qubbas* (green circles) were built amidst tumuli. Images and software: QuickMapServices [35], QGIS 3.4 [36].

settlements [51,52] and monuments patterns [53–55]. A *point pattern* corresponds to a set of locations of spatial events generated by a stochastic process within a bounded region [56]. In this case study, the point pattern corresponds to the *qubbas*' locations and the bounded region is represented by the ROI. The density of a point pattern is proportional to the intensity of the underlying process. The intensity, in turn, can be constant within the region -isotropic- or spatially variable. The first occurrence is called *Homogeneous Poisson Process* (HPP), and corresponds to a stationary and isotropic process. Conversely, an event distribution that is not spatially uniform is called *Inhomogeneous Poisson Process* (IPP). Within an IPP, it is interesting to investigate if the intensity of the events depends on spatial variables (*covariates*) and to quantify such dependence (*First Order Properties*). Furthermore, when analysing a point pattern it may be relevant to assess whether the points show independence from each other or some kind of interpoint dependence occurs (*Second Order Properties*) [57]. The effects of point process intensity (*First Order Properties*) and spatial interaction (*Second Order Properties*) can be hardly distinguishable, although the latter is often recognised at small scale, whereas large-

scale variations are usually associated with the non-stationary intensity of underlying processes [52].

These complementary properties were exploited to investigate the spatial organisation of Eastern Sudan's funerary monuments. PPA has been performed using the package *spatstat* [58] within the software system for statistical computing R [59] through the visual interface of Rstudio [60]. As discussed before, this study aims to assess the impact of the environmental setting of the region on the distribution of the *qubbas* in the regional landscape, and to quantify the spatial clustering of sites. In order to address these research questions, two complementary null hypotheses were tested through PPA:

$H_1$: At landscape-scale, the density of *qubbas* is uniform (the intensity of the point pattern is stationary and isotropic). The spatial variables do not affect the spatial distribution of the points.

$H_2$: At local-scale, the *qubbas* are completely independent from each other presenting no spatial correlation.

In order to statistically address the two hypotheses, the intensity of the point process was explored with nonparametric (see 4.1) and parametric (see 4.2) tools that evaluate the effectiveness of three sets of independent and self-subsistent environmental circumstances (see 4.2.1) potentially ruling the distribution of monuments in the landscape: i) the influence of the topographic position (i.e., elevation, slope, distance from hills and water courses), thus their durability and visibility from mid/long distances; ii) the influence of the availability of in-situ raw construction material, thus the convenience of a reduced cost of transport; iii) the intervisibility between tumuli and *qubbas*, thus some degree of continuity in the choice of location between more ancient and more recent monuments. The sets were chosen according to pragmatic observations about the tombs' locations within the thematic basemap, which allowed an effective preliminary estimation of the PPA's potential results. Additionally, the interpoint dependence of sites has been explored in order to assess the potential existence of an immaterial societal superstructure (see 4.2.2).

# 4 Workflow and results

## 4.1 Nonparametric density estimation

In spatial statistics, the homogeneous intensity $\lambda$ defines the expected number of points per area unit."Homogeneity" in a point process means that the events have no preference for any spatial locations, a condition that is very unlikely to be observed in nature. When the intensity is spatially varying, the number of events per area unit is determined by the function $\lambda(u)$, where $u$ is the location observed in the point process [57]. At naked-eye the general distribution of the *qubbas* shows a clear clustered non-random distribution which could mean that the intensity of the point pattern is inhomogeneous (IPP).

The intensity of the sites was therefore initially explored through a Kernel Density Estimation (KDE), a nonparametric density-based approach that creates a visual representation of intensity showing local anomalies (hotspots and coldspots) from the mean value of $\lambda$ under a standard deviation value (smoothing bandwidth or *sigma*) [58]. KDE was obtained with the *spatstat* function *density.ppp*. The smoothing bandwidth can be specified in the function with the argument *sigma*. The kernel bandwidth *sigma* controls the degree of smoothing: a small value may omit a potential general trend, while a large value may omit local details [53,61]. In this research the sigma was automatically estimated with the *likelihood cross-validation* method in *spatstat* (*bw.ppl*) and manually adjusted through the function argument *adjust*. Several

adjusting values have been tested to select the most visually suitable KDE for this case study (Fig 4).

The default *sigma* does not represent the small clusters of *qubbas*, while the automatic bandwidth estimation (*bw.ppl*) only shows the areas with the highest positive correlation between points. The best representation of the dynamic intensity of the *qubbas* in the ROI's landscape seems to be provided by a manual adjustment of the *sigma* at *adjust = 30*: this KDE adjustment reveals a long-distance linear continuity of sites, yet preserves a clear distinction of at least six large clusters.

## 4.2 Parametric modelling of intensity

The varying spatial intensity of the *qubbas* highlighted by the KDE has been further explored with parametric methods in order to assess whether environmental covariates could explain the inhomogeneous density of the events of the point process. The geomorphological analysis of the region [27] served as a starting point for determining a list of potentially explanatory variables. Topographic characteristics (Elevation, Slope, Aspect, Geomorphon, Topographic Position Index (TPI), Convergence Index (CI)) and resources availability (distances from outcrops of igneous rocks, outcrops of metamorphic rocks and watercourses) of the area were explored to select the best performing set of variables.

**4.2.1 Spatial covariates.** The Elevation values were retrieved from the ALOS World 3D, a global 30-meters resolution (resampled to 50 m resolution) Digital Surface Model (DSM) [62],

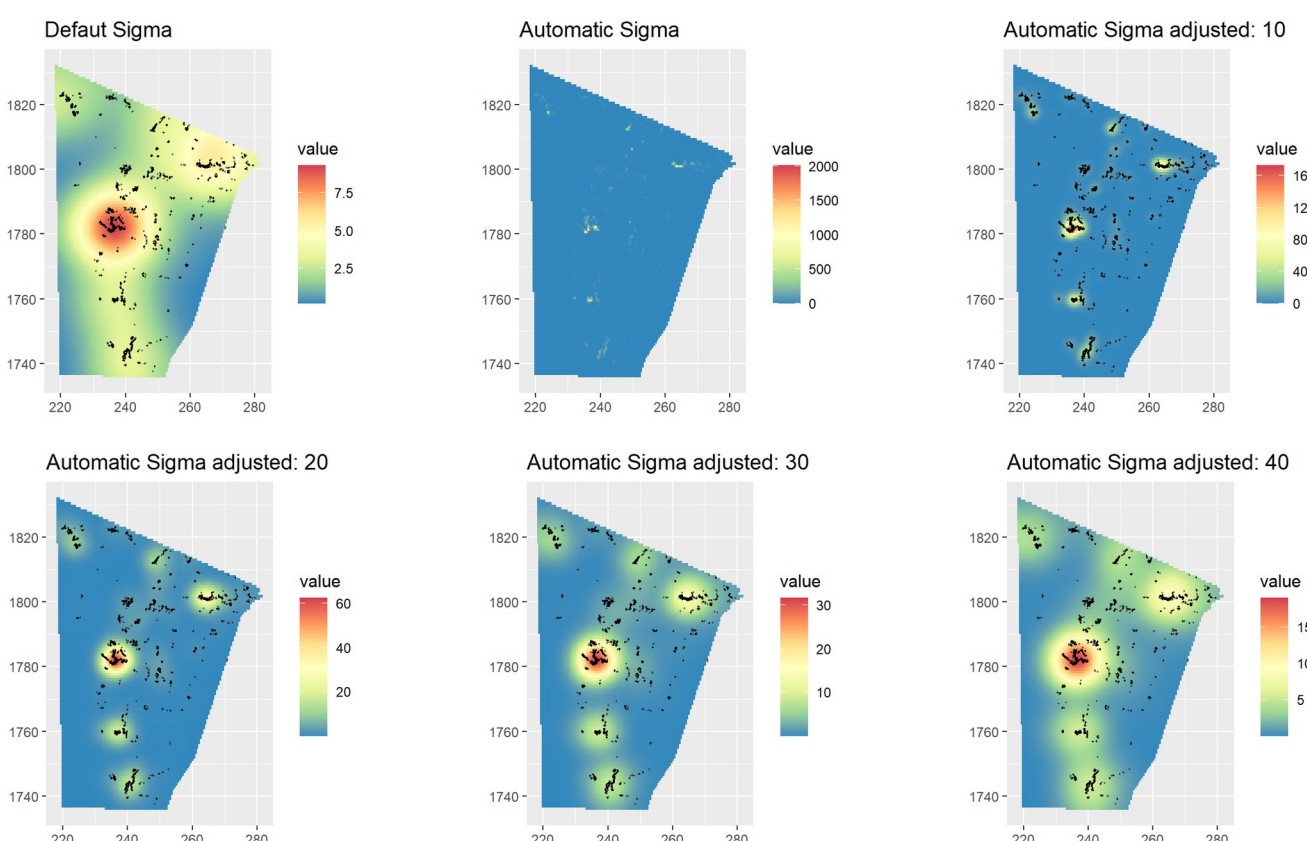

**Fig 4. Kernel density estimated with default, automatic and manually adjusted methods.** From the top left: the default KDE calculated with the *spatstat* function density.ppp.; KDE with sigma automatically estimated with the *spatstat* function *bw.ppl*; the KDE estimated with *ppl sigma* adjusted with values 10, 20, 30 and 40 respectively.

accessed through the Google Earth Engine API [63] with the Python module *geemap* [64]. The DSM was then elaborated in GRASS GIS [65] to extract the Slope and the Aspect covariates with the module *r.slope.aspect* [66]. The Aspect, whose values are defaulted as 0˚-359˚, was split into *Northness* and *Eastness*, linear variables that are more appropriate for subsequent use. Positive *Northness* values correspond to northward orientation, negative values correspond to a southward orientation while values close to 0 indicate either East or West. Similarly, *Eastness* indicates if the orientation is Eastward (positive values) or Westward (negative values) and values close to 0 correspond to either North or South [67]. The covariate *Geomorphon* corresponds to fundamental microstructures of the landscape and it was calculated in GRASS GIS with the module extension *r.geomorphon*. Each geomorphon value (1 to 10) indicates a particular landform: flat area (1), summit (2), ridge (3), shoulder (4), spur (5), slope (6), hollow (7), footslope (8), valley (9), depression (10) [68]. The euclidean distances from lithological parameters (igneous rocks and metamorphic rocks) and from watercourses have been calculated through the GRASS GIS module *r.grow.distance*. Euclidean distance was chosen due to the immediate availability of raw stone at the very hillfoot of the rocky outcrops, a condition that requires no upslope tortuous walking; similarly, the watercourses have no significant depth within the gravelly plains [27]. TPI and CI were employed to measure relative topographic locations such as ridges and depressions. Both methods are independent to absolute elevation values and are extremely effective to highlight topographic prominences within a region. TPI measures the relative topographic position of a cell as the difference between the elevation in that cell and the mean elevation in a predetermined neighbourhood. Positive TPI values (TPI > 0) correspond to hills and ridges while negative values represent locations lower than their surroundings (TPI < 0), such as valleys or pits. TPI values close to 0 correspond to flat areas or persistently inclined slopes [53]. In this research the TPI was calculated with the Topographic Position Index module [69] in SAGA GIS [70] with three different *radii*: 100 m, 500 m and 1000 m. Finally, the CI enables the parameterization of flow convergence and divergence employing the exposition of neighbouring raster cells. This covariate was extracted with the SAGA GIS module Convergence Index [71]: small values correspond to depressions (flow convergence) while high values correspond to ridges (flow divergence) [53]. Lastly, to test a possible influence of the position of the *tumuli* on the location of the islamic tombs, a Cumulative Viewshed Analysis (CVA) has been performed with the GRASS GIS module *r.viewshed.cva* [72]. This module enables the construction of a "*visualscape*" map from a series of input points stored in a vector points map [73], in this case the locations of the *tumuli*. Because visibility is bidirectional, the CVA raster served as a spatial covariate to assess if the *qubbas* have been placed taking in consideration the visibility of more ancient tombs (*tumuli*).

The Pearson correlation's test was employed to test collinearity between the variables, in order to avoid overparameterization in the models (Fig 5).

Collinearity is the high linear relation of two or more predictors. In general, an absolute correlation coefficient of > 0.7 among two or more predictors indicates the presence of collinearity [74]. In this research we decided to consider > 0.6 as a parameter to exclude correlated variables. Elevation shows collinearity especially with the distance from lithological resources. This is explained by the fact that the outcrops emerge from the flat plain with a high slope degree, representing a sudden steep altitude increase. For this reason, Elevation has been excluded from the parametric modelling. Furthermore, TPI 500 and TPI 1000 were removed from the model, and just TPI 100 was maintained because of its overall lower collinearity score. The Pearson's test results also show a high positive correlation between euclidean distances from streams and from both lithologic outcrops. This is due to the tight admixture of metamorphic hills and igneous intrusions, cut by the dense dendritic stream system that originates from the same hillsides where the *qubbas* may be located [27]. Thus, in the model

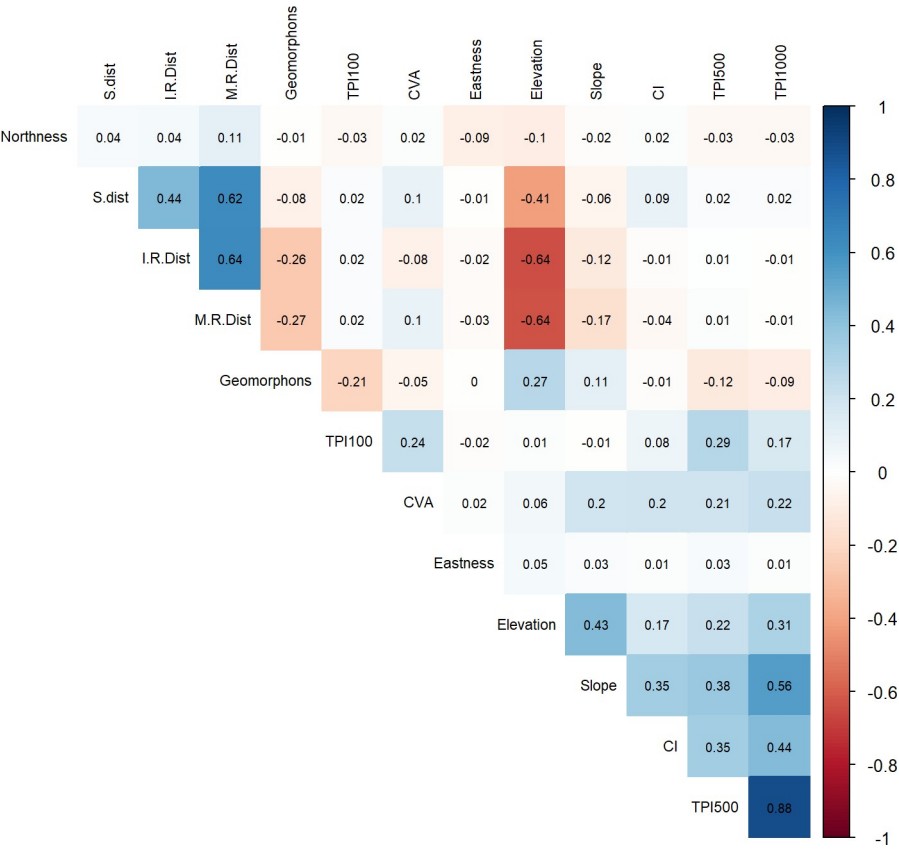

**Fig 5. Pearson correlation's test results.** Collinearity is the high linear relation of two or more predictors. In general, an absolute correlation coefficient of >0.7 (dark blue or dark red) among two or more predictors indicates the presence of Collinearity [74]. In this research we considered > 0.6 as a parameter to exclude correlated variables.

analysis only the euclidean distances from the metamorphic rocks were considered. This choice was driven by the field observation that the *qubbas* are mostly made of naturally occurring flat slabs of foliated metamorphic rocks picked up from weathered hillsides, so it is convenient to only keep the covariate that may better describe how the availability of construction raw material affected the spatial distribution of the points.

The topographic and morphometric variables that show less collinearity (Slope, Northness, Eastness, TPI100, CI and Geomorphon, distance from metamorphic rocks and CVA) have been further investigated to evaluate their actual influence on the point process. The histograms in Fig 6 show univariate relationship between the dependent variable (the *qubbas*) and each covariate.

Both Northness and Eastness seem to have no influence on the landscape spatial distribution of the *qubbas*: the points are uniformly oriented in all cardinal directions. Thus, Northness and Eastness have been excluded from the parametric estimation of intensity. Conversely, the other spatial variables (Slope, TPI 100, CI, Geomorphon) indicate that the *qubbas* are more likely to occur along foothill flat or gently rolling areas. Furthermore, lithological and sociocultural parameters (distance from metamorphic rocks and *Tumuli*'s CVA respectively) seem to have a heterogeneous influence on the spatial distribution of the *qubbas*.

**4.2.2 Point pattern modelling.** The parametric estimation of intensity has been performed by fitting the chosen covariates (Fig 7) to the *qubbas* point pattern with the *spatstat* function *ppm* (Point Process Model) [58]. PPM consists of a range of spatially explicit models

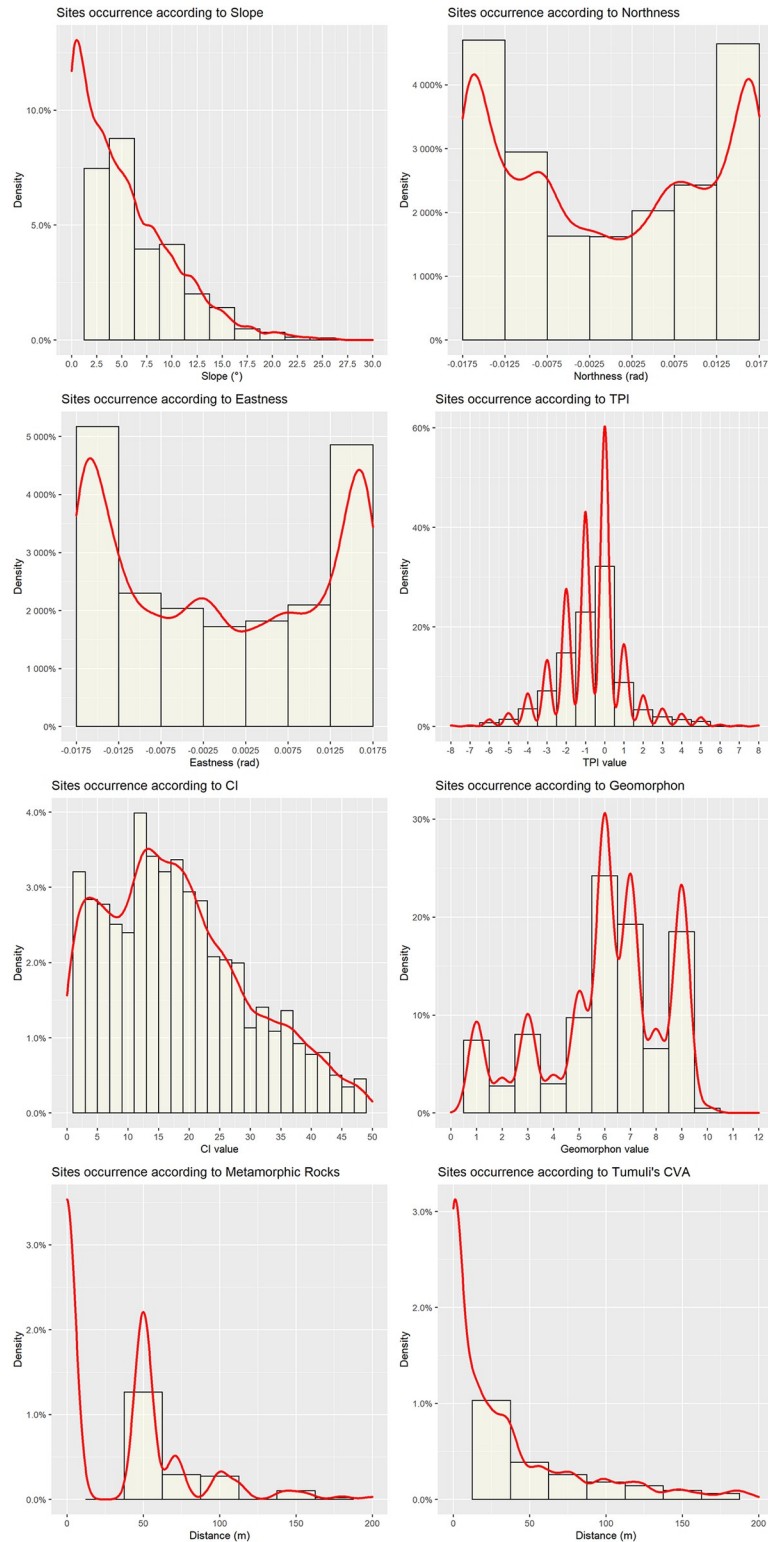

**Fig 6. Spatial variation of the intensity of sites according to covariate values.** Top left to bottom right: Slope, Northness, Eastness, Topographic Position Index 100, Convergence Index, Geomorphon, Distance from metamorphic rocks, Tumuli's Cumulative Viewshed.

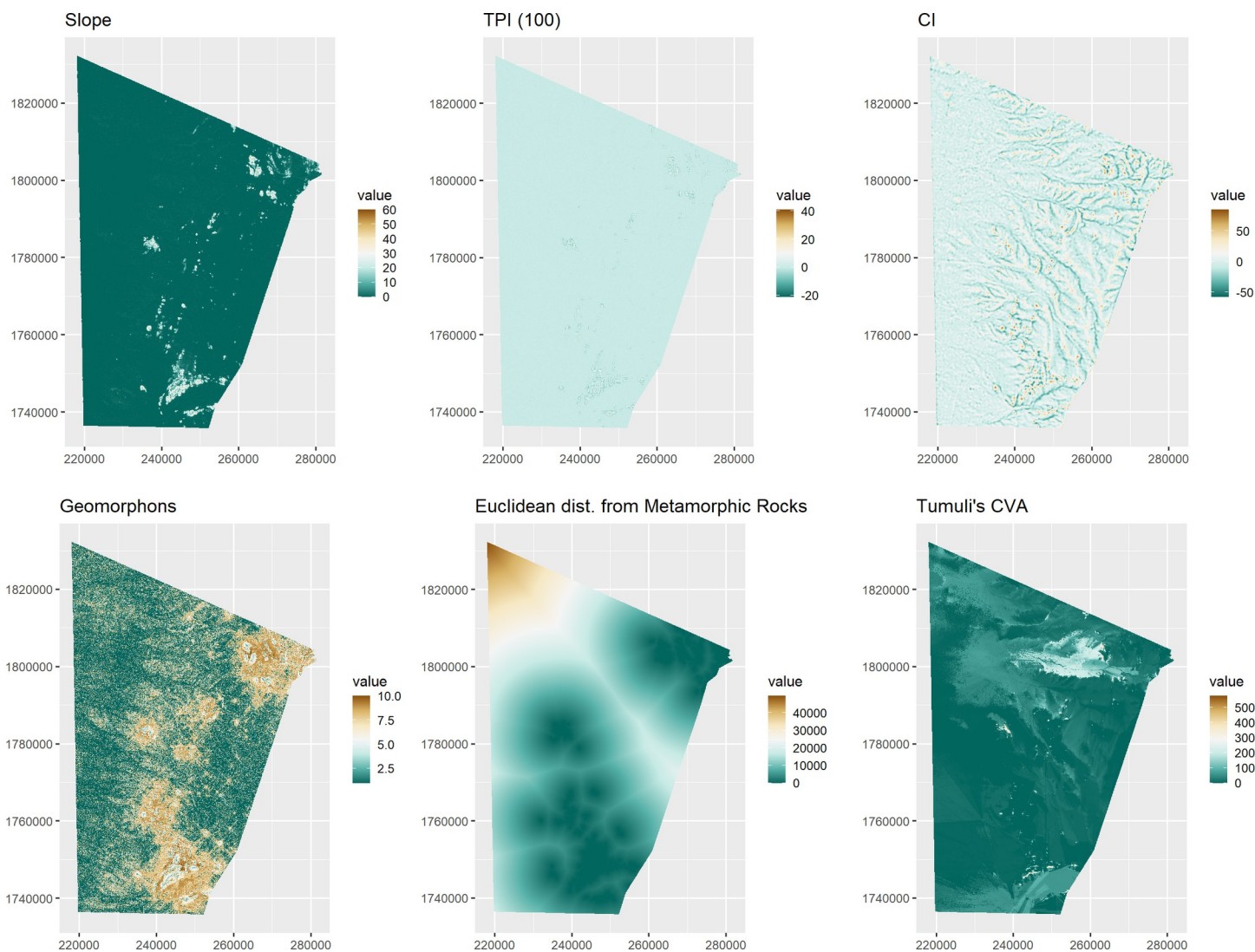

**Fig 7. Chosen covariates.** Top left to bottom right: Slope, Topographic Position Index 100, Convergence Index, Geomorphons, Euclidean distance from metamorphic rocks, Tumuli's Cumulative Viewshed.

that facilitate formal analyses of relationships between point-patterns and spatial covariates [57]. Three different IPP models have been fitted: Model 1 (Topographic covariates), Model 2 (Lithological covariate), Model 3 (Socio-cultural covariate). The Schwarz's Bayesian Information Criterion (BIC) has been employed to compare the competing models and to assess the performance of each covariate [75]. BIC is calculated as the difference between the maximised likelihood of the model and the product of covariates and number of observations (points), therefore a lower BIC corresponds to a better performing model [57]. Following the principle of parsimony, stepwise selection of covariates enables the identification of the combination of variables that minimises BIC values, and the covariates that show no significant correlation with the points are excluded during the process. The remaining covariates were then employed to fit a fourth "hybrid" model that includes the best performing variables (Model 4). The performance of the resulting IPP models are evaluated through the comparison between them

**Table 1. Result of the BIC stepwise covariates selection and model selection based on BIC weights.**

| PPMs | Selected Covariates | Discarded Covariates | BIC | Weights Model 0–1 | Weights Model 0–2 | Weights Model 0–3 | Weights Model 0–4 |
|---|---|---|---|---|---|---|---|
| 0 | - | - | 285618.727843511 | 0 | 0 | 0 | 0 |
| 1 | *Slope, TPI100, CI, Geomorphon* | None | 267576.992701805 | 1 | 1 | 1 | 0 |
| 2 | *Eu. Dist. Metamorphic Rocks* | None | 279957.068568553 | - | 0 | 0 | 0 |
| 3 | *Tumuli's CVA* | None | 284537.742730204 | - | - | 0 | 0 |
| 4 | *Slope, TPI100, CI, Geomorphon, Eu. Dist. Metamorphic Rocks, Tumuli's CVA* | None | 266045.103883927 | - | - | - | 1 |

0 and 1 values in the *Weights* columns to be read as *loser* and *winner* respectively.

and with an alternative model: Model 0. This is a HPP model created by fitting a constant value to the point patterns. If the investigated sites are uniformly distributed in the study region and the spatial variables do not influence their locational pattern ($H_1$) (see 2.2), Model 0 is expected to overperform the other Models. BIC weights, instead, are employed to provide a normalised estimation of the relative performance of the five models in order to assess which one better explains the spatial distribution of the observed points [76].

BIC and stepwise model selection can be performed with the R package *MASS* [77] function *stepAIC* (Akaike Information Criterion) using the argument $k = log(n)$ where $n$ is the multiple of the number of degrees of freedom used for the penalty.

The significance of BIC-selected covariates has been validated against the stationary model (Model 0), using BIC weights. The ppm fitted with topographic variables (Model 1) scored the best performance against the stationary model (Model 0) and against the non-stationary models fitted with the distances from construction material (Model 2) and the intervisibility with tumuli (Model 3) respectively. Nevertheless, during the stepwise selection no covariates have been excluded in Models 1, 2 and 3, meaning that all the spatial variables show significant correlation with the points (Table 1). Thus, the hybrid model (Model 4) has been fitted using all the six initially selected covariates (Fig 7) and it scored the lowest BIC value (Table 1). In other words, the hybrid model (Model 4) explains the IPP underlying site distribution more efficiently than all the other models (0,1,2,3).

The coefficients of the hybrid model (Model 4) show an inverse correlation of the sites' locational pattern with Slope, TPI and the distances from metamorphic rocks (Eu. Dist. Meta). Conversely, the coefficients of BIC-selected Model 4 show a direct correlation with CI, Geomorphon and CVA (Table 2). This result suggests that the selected environmental covariates had a significant influence on the spatial distribution of the sites. Thus, these results show that point process intensity is clearly inhomogeneous, and the first null hypothesis ($H_1$) can be rejected.

**Table 2. Covariates of the BIC-selected Model 4.**

| Covariates | Estimate | S.E. | CI 95% lo | CI 95% hi | Z test | Correlation |
|---|---|---|---|---|---|---|
| Slope | -4.492999e-03 | 1.449379e-03 | -7.333730e-03 | -1.652268e-03 | <0.001 | Inverse |
| CI | 5.986409e-02 | 6.500641e-04 | 5.858999e-02 | 6.113819e-02 | <0.001 | Direct |
| TPI | -1.309407e-01 | 4.892103e-03 | 1.405291e-01 | -1.213524e-01 | <0.001 | Inverse |
| Geomorphon | 1.695567e-01 | 3.900717e-03 | 1.619115e-01 | 1.772020e-01 | <0.001 | Direct |
| Eu. Dist. Meta | -6.269355e-05 | 1.828274e-06 | -6.627690e-05 | -5.911020e-05 | <0.001 | Inverse |
| CVA | 1.433004e-03 | 1.617823e-04 | 1.115916e-03 | 1.750091e-03 | <0.001 | Direct |

The rejection of $H_1$ proves that the sites' distribution pattern within the ROI is driven by landscape and environmental preferences. In order to investigate the possibility that additional, undetected causes may have affected the spatial distribution of the *qubbas*, the interpoint dependence of sites ($H_2$) has been assessed applying the L-function [78] to Model 4.

The Besag's L-function is a commonly used transformation of the Ripley's K-function [79]. The Univariate [80] and the Bivariate [52,81] K-functions have been widely applied in archaeological studies. The L-function transforms the theoretical Poisson K-function $K_{pois}(r) = \pi r^2$ to a straight line $L_{pois}(r) = r$, easing the visual assessment of the resulting graph [57]. The L-function measures the number of events at different distances from any event and, in this paper, it has been employed to explore the clustering of the *qubbas* in the study area ($H_2$). The L-function can be calculated in *spatstat* with the functions *Lest* [58] and in this research it was calculated from the observed sites, as well as for 999 Monte-Carlo simulated point patterns. In order to visually estimate the impact of first-order properties of the point patterns on their second-order properties, the L-function of Model 0 has also been computed (Fig 8). The observed values completely exceed the 95th percentile of simulated values (the highest limit of the confidence envelope) both in the homogeneous (Model 0) and inhomogeneous (Model 4) plots, suggesting an extremely high positive correlation of points that cannot be explained with landscape preferences alone.

Because the *qubbas*' spatial distribution appears to likely be affected by unobserved/undetectable random influences, a cluster point process [57] was strategically employed to examine their arrangement. In spatial statistics, cluster processes are modifications of the Poisson process to incorporate additional random influences (doubly stochastic Poisson process) [82], and are often chosen when the point pattern is likely to be influenced by unobserved covariates.

In this research, a *Neyman-Scott* Cluster process [83] was employed. Neyman Scott Cluster (NSC) processes are foremost models for clustering point patterns of objects' locations. This spatial statistic tool was developed to study the spatial pattern of galaxies and is widely applied in forestry ecological studies [84]. This case study represents its first application in a (geo) archaeological context. The NSC model can be seen as the result of a two-stage random mechanism: 1) unobservable *parent* points are generated and located according to a HPP; 2) a random amount of *offspring* points is generated by each parent point and scattered around their locations. The offsprings are the actually observed points and their locations depend on their parents' locations. In *spatstat* NSC model can be fitted to a point pattern data by the function *kppm* [58]. In this study a particular type of the NSC process called *Thomas cluster process* [85] was employed, using the argument *method = Thomas* in the *kppm* function.

This choice was driven by the KDE's glaring display of dense and discrete hot spots (Fig 4), a condition that suggests the presence of regions of high intensity around "*parent*" points against an empty background. Furthermore, the nearest neighbour (nn) distances indicate that the *qubbas* are very close (min nn distance: 0,72 m; mean nn distance: 23,3 m) so the occurrence of a positive dependence between points was tested.

Two different *kppm* have been fitted: a stationary NSC model (Model 5) and an inhomogeneous NSC model (Model 6) resulted by fitting the point process intensity to the set of covariates of the *ppm* with the best BIC score (Model 4). Finally, $H_2$ has been addressed applying the L-function to Models 5 and 6 and compared to the results of Model 0 and 4 respectively (Fig 8).

The results have been plotted together in order to visually estimate the impact of first-order properties of the point patterns on second-order properties. Fig 8 displays clearly that the jointly fitted first and second order factors are able to account for the observed point pattern. The results display no deviations of the observed values from the confidence envelope. This suggests that the distribution of *qubbas* can be accounted for by broad landscape preferences (Table 2) with local tendencies towards tomb clustering. In other words, Model 6 best explains

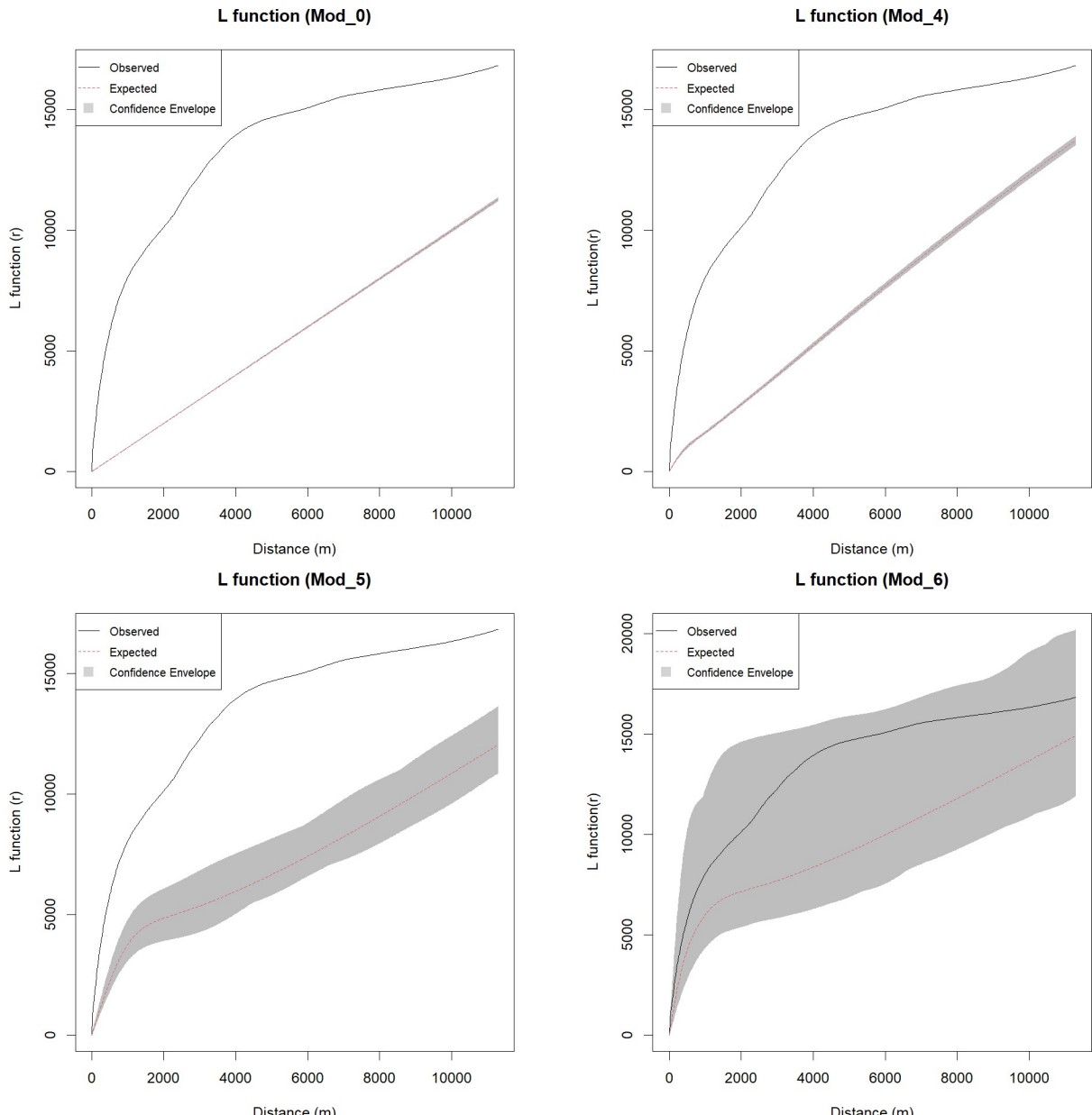

**Fig 8. *L-function*s results of homogeneous and inhomogeneous models.** These highlight the influence of spatial covariates on the *qubbas* general spatial trend.

the spatial distribution of the *qubbas* in the ROI and $H_2$ can be rejected: at local scale the tombs are strongly correlated and organised in galaxy-like clusters. Like stars revolving in hierarchical gravitational systems, hundreds of offspring burials are attracted by a few undefined parents serving as invisible centres of gravity, which in turn aggregate according to environmental constraints. The fitted NSC models thus revealed an 'undetectable random influence' that could be interpreted as the results of socio-cultural choices. The parameters of the two NSC process models show that the *qubbas* spatial clustering is characterised by a very low number of parent points and a mean cluster radius of $\approx 475$ m (Table 3). That is, the *qubbas* are distributed among few, very large cemeteries that show no obvious initial (*parent*) tombs

**Table 3. Parameters of the Stationary cluster point process Model 5 and of the Inhomogeneous cluster point process Model 6.**

| Models | Intensity | Kappa (Intensity of the parents) | Scale (cluster radius) | Mean cluster size (number of offsprings) |
|---|---|---|---|---|
| Kppm 5 | 2.505782e-06 | 1.606992e-08 | 4.815851e+02 | 155.5071 points |
| Kppm 6 | *Log intensity: ~Slope + CI + TPI + Geo + Meta + CVA* | 1.66968e-08 | 4.72056e+02 | *Mean cluster size: [pixel image]* |

yet drew continuous attention possibly among different families/groups and over several generations.

## 5 Discussion

### 5.1 Effectiveness of the PPA workflow

The results of the PPA analysis stimulate relevant observations about both First and Second Order Properties of the *qubbas*' spatial distribution.

Firstly, the coefficients of the Model 4 (Table 2) demonstrate that the *qubbas* are more likely to occur along foothills flat areas with higher availability of raw material and mutual visibility with the ancient tumuli ($H_1$ rejected). The northwesternmost clusters featured in the ROI seem to represent a partial exception to the Model, because despite sitting on flat raised hillocks, satellite images show no glaring source of raw stone material. A targeted field survey may help to reveal whether they belong to a completely different construction style or if the stones were gathered from a different source, possibly a hamada desert pavement or a residual surface [27] resulted from the pre-Quaternary and Quaternary weathering.

In order to verify the Model's goodness-of-fit, the measuring process of the residual values has been carried out. Residuals are obtained by subtracting the fitted intensity function from the observed counts for each region of the model quadrature. By plotting the smoothed version of residual for the whole area, spatial variability in model prediction can be visually assessed: positive residuals occur where the model underestimates the true intensity (i.e. site density), negative residuals occur where the model overestimates the true intensity [57]. The diagnostic smoothed residual values of the Model 6 have been calculated through the *spatstat* function *residuals.kppm* [58]. As shown in Fig 9, the smoothed residual values of Model 6 display an overall uniform level of prediction, except for a limited portion in the centre of the ROI in which the model greatly underestimated the true intensity of sites. The area corresponds to a small outcrop where the remote survey documented a higher occurrence of *qubbas* than in the surroundings ($\sim$1/3rd of the dataset). This significant positive autocorrelation (similar values are close to each other) [86] might imply that the location became a pole of attraction of disproportionate size against the others in the region. The causes of this growth may abide by a positive feedback that was originally triggered by exceptionally advantageous environmental settings and subsequently perpetuated by arbitrary choices driven by individual kinship and collective social memory ($H_2$ rejected).

Likely, social memory-driven choices may also emerge assessing the possibility of a cultural continuity between tumuli and *qubbas*. In fact, the CVA coefficient of Model 4 shows that there is a slight direct correlation between the distribution of *qubbas* and the intervisibility with *tumuli* (Table 2). In order to further explore the spatial correlation between the two categories of monuments, the cross-type nearest-neighbour function $G_{ij}(r)$ has been calculated. The estimation of the cross G function is based on measuring the distances between each point of type *i* (*qubbas*) and the nearest point of type *j* (tumuli), and it is computed in *spatstat* by the function *Gcross* [58]. The resulting graph (Fig 10) shows that for great distances (>400m) the

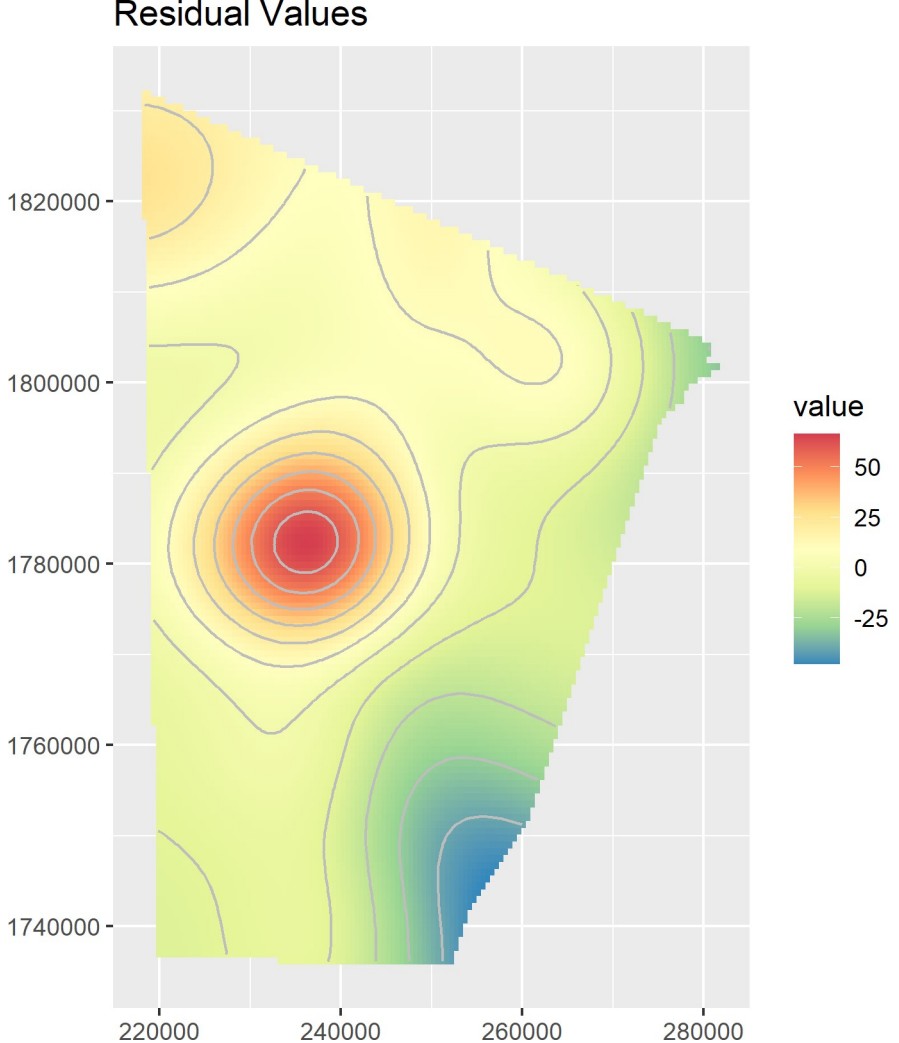

**Fig 9. Residual values.** Positive residuals (red hue) occur where the model underestimates the true intensity (i.e. site density), negative residuals (blue hue) occur where the model overestimates the true intensity [57].

cross G-function displays a segregation pattern that is probably related to the dispersal of the *qubbas* in the ROI towards locations where the tumuli are not found. On the contrary, at shorter distances (<400m) there is an aggregation of the *qubbas* around the tumuli that is consistent throughout the region.

The trend towards an aggregation pattern well fits into a two-fold interpretation that contemplates variable degrees of environmentally constrained solutions and complex social dynamics. Nevertheless, since *"correlation does not mean causation"* [57], several considerations upon cultural continuity based upon the available archaeological and anthropological data are required.

## 5.2 The funerary landscape

Our investigation highlighted that the *qubbas* were built in extremely large numbers up to more than 60 km south of the site of Jebel Maman, thought to be the southernmost occurrence of Beja Islamic funerary monuments [10,23–25].

## Cross G function

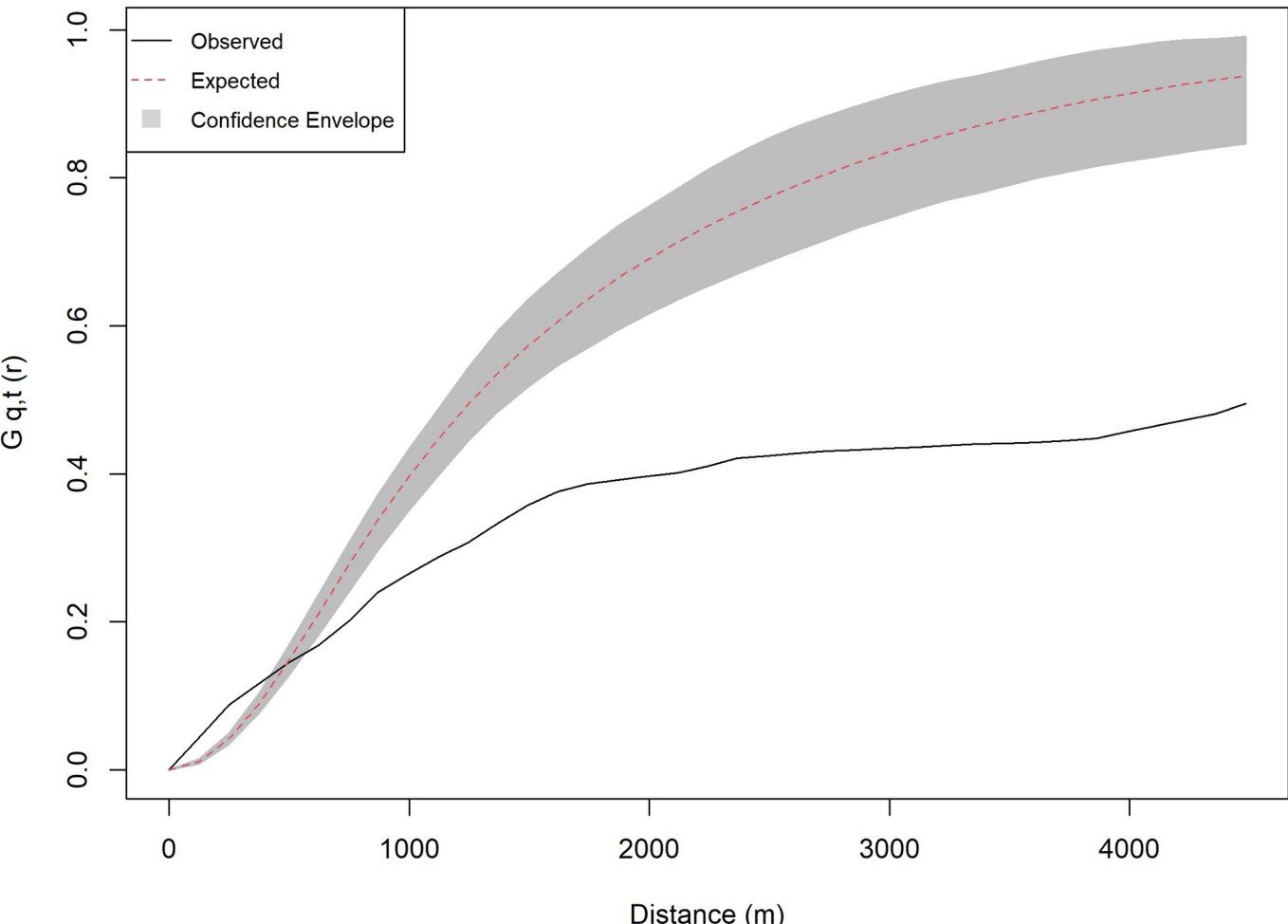

**Fig 10. Results of the cross G-function.** *The* estimation of the cross G-function is based on measuring the distances between each point of type i (*qubbas*) and the nearest point of type j (*tumuli*), and it is computed in *spatstat* by the function *Gcross* [57]. Where the *observed* values exceed the *expected* values (<400m), an aggregation pattern in revealed.

Scholars [23–25] drastically underestimated the amount of *qubbas* dotting the region, confirming the essential role of remote sensing when dealing with landscape-wide cultural processes and more in general the complexity of the funerary landscape. The PPA analysis suggests that environmental features are the triggering factors in the selection of locations for funerary monuments, thus representing a typical example of geological opportunism [87]. Yet, at the local scale, socio-cultural factors must be considered for justifying the tight clustering of monuments. Such factors may reside within lineage dynamics of the Beja people and the presence in the surroundings of ancient tumuli that were orally handed down as belonging to some bygone ancestors. In fact, Beja social memory suggests that they settled the region since "time immemorial" [46], and ancient written sources mention their existence well before 2000 years ago. Unfortunately, at present this cannot be thoroughly assessed lacking a comprehensive ethnoarchaeological survey of Beja culture. Nevertheless, a complex formation of the funerary landscape as result of cultural behaviours is evident. This means that the repeated and well-coded use of certain locations for Islamic cemeteries traces back to more ancient local

cultural tradition and to the identification of suitable places for tumuli and the concomitant opportunistic exploitation of occurring raw building material. This resulted in a diachronic stratification of several architectural styles of monuments that portrays how the funerary habits of a millennia-old society persisted almost undisturbed, preserving the value of location and kinship despite external religious and stylistic influences. It is yet to be determined whether the larger identified necropoles grew on local population intake alone or used to draw interest at supra-regional scale, as neither archaeological surveying of living structures nor thorough studies on ancient population density have been carried out as of today. The very city of Kassala was founded only in the 1830's during a time of foreign incursions and conflict with the Turks and Egyptian governors of Suakin (present days Red Sea State of Sudan), which ultimately resulted in the launching of the first cotton cultivation scheme in the Gash delta in the 1860's, representing the first substantial disruption of the natural ecosystem and the first serious change in lifestyle and land use pattern for the otherwise sparsely settled semi-nomadic Beja cattle breeders [46], which may have left little to no evidence of their transit.

## 6 Conclusions: "cosmogony" of burials

Here we present the first application to archaeology of a spatial statistics tool that was developed for cosmology. The NSC revealed "invisible", empirically undetected local-scale clusters of Islamic funerary monuments that were nested within six large, environmentally driven "visible" clusters outlined within the study region. Considering such monuments as realisations of galaxy-like clusters composed of offspring points revolving around parents, enabled a better understanding of their spatial organisation, regardless of their chronology and shape. In fact, one of the premises of this research was the almost complete lack of previous studies on their origin, precise chronology and stylistic variants. With this approach we were able to formulate new hypotheses about their genesis and role within the funerary landscape. Comparing their arrangement with historical sources that describe the groups inhabiting the area since the late 1st millennium BCE, it emerges that the local clusters are most probably tribal/family cemeteries of the Beja people, where the socially superstructural offspring burials grew around environmentally opportunistic parent tombs, built in favourable locations with readily available construction material. Lastly, the cross G-function shows a local-scale (<400m radius) tendency towards aggregation of the Islamic funerary monuments around much older ones belonging to ancient pan-African traditions. This aspect requires further insights to determine the ratio of societal and geologically opportunistic contributions to their locations, yet it highlights continuity and stratification of subsequent phases.

The application of NSC to our archaeological data disclosed a galaxy-like distribution of funerary monuments and suggested the existence of elusive factors affecting the spatial distribution of cemeteries. In monumental funerary archaeology and other archaeological contexts, especially those in remote desert areas where features extend beyond reach and fieldwork opportunities are limited, NSC paired with remote sensing represents an invaluable, fresh tool for answering relevant questions on the origin and development of archaeological sites, landscapes and palimpsests.

## Acknowledgments

This research was carried out as part of S.C.'s PhD project within the frame of the IAEES—Italian Archaeological Expedition to the Eastern Sudan, University of Naples "L'Orientale"– ISMEO. Field pictures are part of the archives of the Italian Archaeological Expedition to the Eastern Sudan. We thank the archaeological authorities of the Republic of the Sudan, the National Corporation for Antiquities and Museums, the Ministry division in Kassala, their

welcoming staff and the skilled drivers who made the field surveys possible. We thank Francesco Carrer (Newcastle University, Newcastle upon Tyne, UK) for his comments on the R script code, and Isaac Ullah (San Diego State University, San Diego, CA—USA) for his suggestions on the use of the Cumulative Viewshed Analysis Grass GIS module. Finally, we wish to thank the reviewers Michael Kempf and Julia Budka for their feedback and contribution to the final version of this paper.

## Author Contributions

**Conceptualization:** Stefano Costanzo, Filippo Brandolini.

**Data curation:** Stefano Costanzo, Filippo Brandolini.

**Formal analysis:** Filippo Brandolini.

**Funding acquisition:** Andrea Zerboni, Andrea Manzo.

**Investigation:** Stefano Costanzo, Habab Idriss Ahmed, Andrea Manzo.

**Methodology:** Filippo Brandolini.

**Resources:** Andrea Manzo.

**Software:** Filippo Brandolini.

**Supervision:** Andrea Zerboni, Andrea Manzo.

**Visualization:** Stefano Costanzo, Filippo Brandolini.

**Writing – original draft:** Stefano Costanzo, Filippo Brandolini, Andrea Zerboni.

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
