## [Decision Letter · Decision Letter 0]

5 May 2021

PONE-D-21-07238

Creating the funerary landscape of Eastern Sudan

PLOS ONE

Dear Dr. Costanzo,

Thank you for submitting your manuscript to PLOS ONE. After careful consideration, we feel that it has merit but does not fully meet PLOS ONE’s publication criteria as it currently stands. Therefore, we invite you to submit a revised version of the manuscript that addresses the points raised during the review process.

Both reviewers have a number of small comments and suggestions, which I think will improve the paper. I would particularly encourage some editing to unify the styles used (possibly by different co-authors) in different sections, and to distinguish descriptions of workflow from results, as pointed out by Reviewer 1. I also think it would be good to clarify the use of 'galaxy-like' in describing both a statistical method borrowed from cosmology *and*, at the same time some behavioral implication of the clustering of funerary structures (as pointed out by Reviewer 2). Reviewer 2 also recommends some recently published literature to cite. 

We look forward to receiving your revised manuscript.

Kind regards,

Radu Iovita

Academic Editor

PLOS ONE

Journal Requirements:

[The IAEES is funded by the Universitàdi Napoli “L’Orientale”, ISMEO -Associazione Internazionale di Studi sul Mediterraneo e l’Oriente, and the Italian Ministry of Foreign Affairs. The Expedition is also generously supported by the Regional Government of the Kassala State. Additional financial support was provided by Italian Ministry of Education, University, and Research (MIUR) through the project ‘Dipartimenti di Eccellenza 2018–2022’ (WP4—Risorse del Patrimonio Culturale) awarded to the Dipartimento di Scienze della Terra ‘A. Desio’ (University of Milan, Italy).]

 [The author(s) received no specific funding for this work.]

4. We note that Figures 1, 4, 7 and 9 in your submission contain map/satellite images which may be copyrighted. All PLOS content is published under the Creative Commons Attribution License (CC BY 4.0), which means that the manuscript, images, and Supporting Information files will be freely available online, and any third party is permitted to access, download, copy, distribute, and use these materials in any way, even commercially, with proper attribution. For these reasons, we cannot publish previously copyrighted maps or satellite images created using proprietary data, such as Google software (Google Maps, Street View, and Earth). For more information, see our copyright guidelines: http://journals.plos.org/plosone/s/licenses-and-copyright.

You may seek permission from the original copyright holder of Figures 1, 4, 7 and 9 to publish the content specifically under the CC BY 4.0 license. 

If you are unable to obtain permission from the original copyright holder to publish these figures under the CC BY 4.0 license or if the copyright holder’s requirements are incompatible with the CC BY 4.0 license, please either i) remove the figure or ii) supply a replacement figure that complies with the CC BY 4.0 license. Please check copyright information on all replacement figures and update the figure caption with source information. If applicable, please specify in the figure caption text when a figure is similar but not identical to the original image and is therefore for illustrative purposes only.

Reviewers' comments:

Reviewer's Responses to Questions

**Comments to the Author**

1. Is the manuscript technically sound, and do the data support the conclusions?

Reviewer #1: Yes

Reviewer #2: Yes

2. Has the statistical analysis been performed appropriately and rigorously? 

Reviewer #1: Yes

Reviewer #2: Yes

3. Have the authors made all data underlying the findings in their manuscript fully available?

Reviewer #1: Yes

Reviewer #2: Yes

4. Is the manuscript presented in an intelligible fashion and written in standard English?

Reviewer #1: Yes

Reviewer #2: Yes

5. Review Comments to the Author

Reviewer #1: Review of: PONE-D-21-07238, entitled "Creating the funerary landscape of Eastern Sudan".

Stefano Costanzo and colleagues provide an innovative approach to analyze, visualize, and interpret the archaeological remains of tumuli and tombs in eastern Sudan. The research approach of the authors includes point pattern analysis and the integration of an underlying explanatory covariate using quantitative statistics (R software) and spatial modelling (GIS). The applied methods provide a highly useful tool to model point patterns and to understand the role of environmental explanatory parameters like resource availability, terrain roughness, or other variables. In the paper, the authors aim at tracing the spatial patterns of (visible) burial places as a function of visibility and resource availability/accessibility, which is basically a pragmatic reaction to the distribution of the available raw material in the region. Furthermore, they integrate basic remote sensing techniques, mostly based on satellite imagery to identify and interpret visible archaeological remains.

In general, this article provides a very good overview of the methodical workflow of point pattern analysis and furthermore integrates a new parameter to spatial modelling and statistical methods in landscape archaeology (Neyman-Scott Cluster Process), which allows for the detection of ‘nested clustered’ spatial patterns within larger clusters. That can be considered an interesting and innovative approach to quantitative methods in general and particularly the evaluation of large archaeological datasets. I would also like to particularly emphasize the author’s attempt to not just simply map and plot but to include also socio-cultural behavior, perception, and group identity performance into the modelling outlook of their research. Consequently, this paper is an informative contribution to archaeological research in general and I am looking forward to seeing this article published.

However, I would like to discuss some minor points, mostly concerning the structure of the paper (for individual comments, please see below). The paper clearly is a methodical paper, which offers the reader an introduction into current approaches in spatial statistics and modelling techniques (see also the previous research of the authors). For this reason, the actual structure of the paper follows a short introduction into the archaeological and environmental settings, followed by a Material and Methods section and a Workflow and Results paragraph. The introduction would benefit from some reconsiderations about the main goal of the paper. Correct me if I am wrong, but it looks a bit like the archaeological settings are used as canvas to apply in-depth statistical methods. I also have the feeling that some parts of the introduction and some parts of section 2 (mostly archaeology) were written by a different author compared to the rest of the paper (see also stylistic and spelling errors, -> comments below). The environmental section is very well written and provides a sufficient overview of the study area’s properties.

The following section 3 (Material and Methods) provides a short overview of the data acquisition and some outlines of point pattern analysis (PPA). This section is followed by paragraph 4, which describes the workflow and also the results of the analysis. That s a tricky thing and could meet with opposition regarding the structure of the paper. From my individual point of view, I am totally convinced that it is not easy to distinguish the methodical workflow from the results because every single step in the analysis provides new potential results, which then require adjusted methods. The workflow itself can be considered an important part of the results and thus cannot function as a stand-alone section – it would give the impression of a completely detached section and decrease the readability of the paper (or any other paper). On the other side, the authors do integrate a Material and Methods paragraph, which (methodically) renders the results section a mere results section. If the authors can find a smooth way to overcome these structural issues, the overall outlook of the paper would be even more convincing.

General comments (please consider that I am not a native speaker)

Abstract:

41: behavior

46: to decipher

51: maybe introduce cairns as funerary elements earlier to a broader readership of PLOS One

52: which is still embedded. Do you mean today?

1 Introduction:

57: extremely?

60: intricate? Maybe complex or nested?

61: missing reference

62: vegetation cover?

68: usually instead of oftentimes?

67-70: consider rewording this section

73: infrastructure

73: subject to, prone to, vulnerable?

74: as much as more eminent? Like many other?

75: One example is the hilly Kassala region and Eritrean borderland in Eastern Sudan, where…

76: copius scatters? Maybe scattered patterns?

78: that such monuments occur in forms belonging… please consider rewording this phrase

80: missing word. And classified?

Also 80: around the shared characteristic of being round… that sounds odd. Consider rewording.

81: are of instead of have?

82: Another

85: with early authors you mean: Previously, or previous authors, or previous results..

85: please reconsider the wording of ‘tribes’, which has a negative connotation. Maybe ‘groups’ is more elegant or just “Beja people”.

89: , which led to the..

95: what is a traditional GIS approach?

2 Geography and Archaeology

The first part is generally well written and provides a very good overview of the study area!

109: are located, and delete respectively

116: erosive power? What means alternance?

117: extant? You mean the modern? I mean, you are using modern elevation data so I assume modern would be the right word.

120: Archaeological records from the… suggest…

122: particularly

122: Mesolithic versus Mesolithic and Palaeolithic, be concise in the paper.

124: refer to the original references here!

125: what do you mean with relatively sedentary? Reconsider this methodologically.. ‘relatively’ is not a very precise expression in scientific work..

136: in line 80 you state that they share round characteristics! This is contradictory to what you say here…

140: According to previous results (citation…)

141: they not there

142: Two-storey domed

142-144: maybe use century and not secolo or sec. maybe use 16th century AD -> check the journal guidelines for this.

144-145: Their spatial extent stretches from… to or: They stretch from… to

145: in the north

144-147: this is grammatically wrong and hard to understand. Please rephrase.

151: satellite views is certainly not correct. Imagery or images

3 Material and Methods

This is very well written and summarizes nicely current approaches in spatial analyses and quantitative methods. It shows that the author(s) (maybe the work was split because this paragraph shows differences in linguistic style) are very familiar with the statistic approaches presented in this paper (and in general). For a broader readership and to facilitate the reproducibility of the work, it would be helpful to add some more references to this part.

4 Workflow and results

In this paragraph, the authors are mixing up methodical approaches and subsequent results. I know, it is – particularly when applying a statistical approach – not easy to distinguish into a strictly methodical part and a results/discussion section because the workflow itself can be considered a result per se. In this case, paragraph 4 is a methodical outline of the paper and represents the core of the research. The paper thus turns into a completely methodical and methodological article – which is a good thing! – but this can also be more explicitly highlighted in the abstract and the introduction.

216: A Kernel Density…

216-219: please add references

242/258/9: igneous vs Igneous and metamorphic. Be concise.

357 add model before hybrid

366: site’s ? the site’s distribution? Better: site distribution patterns?

369 and others: L-function, K-function instead of L function. Please be concise

379: add the before L-function

378-379: why do you think so? Spatial interaction only at small scale? If you just state this without adding a discussion, please cite. However, I am not totally sure that this is actually true…. I know that in quantitative methods, a radius or sigma needs to be determined, which is a totally subjective perspective but there are other many authors applying different radii in their research. This is also dependent on terrain permeability and accessibility and hence needs to be reconsidered – at least with a (small) literature review…

391: its instead of their

437: to reveal. Delete again, delete indeed, exists, resulting, weathering processes. Consider also rewording this whole sentence.. maybe split it in two?

490: again: tribes. Please use groups

Figures:

Fig 1: Consider changing the size and the colors of the archaeological features in the map to increase readability (although I am aware that the manuscript will be published online).

Fig. 3: please add north arrow and a grid

Reviewer #2: The results of the study are relevant observations about the funeraryscape of Eastern Sudan and of the qubbas' spatial distribution. This is an important paper with fresh material and new approaches regarding the assessment of burialscapes in Sudan and elsewhere, highlighting the potential of advanced spatial statistics.

This important paper should clearly be published; I suggest that some minor points are reconsidered and maybe explained in more detail.

Here are some brief comments on terms and content:

Line 94 and passim: Could you elaborate why „monumental funerary landscape of Eastern Sudan“ is an appropriate term? Why and how are these tombs monumental? What is your reference point here, thinking of other monumental stone monuments etc.

Line 100: Maybe you want to refer for the „formation of funerary landscaped in arid regions“ to recent work in the Fourth Cataract or in the Bayuda? See below

Line 416: Since you employed the NSC models, I completely understand that it was tempting to use „galaxy-like clusters“ for your tomb settings – but could you maybe explain what „galaxy-like“ really means here? The same holds true for line 511 “cosmogony” – as much as I like this, is it really an appropriate term for the burialscape of Eastern Sudan? If so, why?

Line 417: Could you elaborate what the precise socio-cultural choices were? In what respect are they “precise”? See also your conclusion where a number of questions are still left open and will only be answered by fieldwork.

Line 498ff: Are there any means to explain the comparably very low number of tumuli compared to qubbas? How those this difference in quantities affect or not affect your results and proposed narrative of a continuous use/occupation?

Some more Sudan-specific references for GIS and geoarchaeological projects could be useful, thus potential/optional additions to references 2,3 and 4,5:

Beuger, André 2018. The Geoarchaeology Web Service 2.0: open archaeological geodata of the Bayuda. In Lohwasser, Angelika, Tim Karberg, and Johannes Auenmüller (eds), Bayuda studies: proceedings of the first international conference on the archaeology of the Bayuda Desert in Sudan, 141-156. Wiesbaden: Harrassowitz.

Abdalla Jadain, Modather 2018. An archaeological survey north of the 6th Cataract: GIS study of Sabaloka East. Der Antike Sudan. Mitteilungen der Sudanarchäologischen Gesellschaft zu Berlin 29, 39-57.

For note 5, a recent monograph has appeared:

Jiménez-Higueras, Ángeles 2020. The sacred landscape of Dra Abu el-Naga during the New Kingdom: people making landscape making people. Culture and History of the Ancient Near East 113. Leiden; Boston: Brill

Note 6 is of course the seminal study of one of the authors, but maybe references to other areas of the Sudan could be made? For example, the Fourth Catarct region was labelled as "one continuous cemetery of buried civilisations” (Jackson 1926: 1 = Jackson, H. C. (1926): A trek in the Abu Hamed District, Sudan Notes and Records 9, No. 2, 1–35, see Budka 2006, in: MittSAG, Der Antike Sudan), and this statement was reassessed by modern surveys in the 2000s, see especially works by Welsby, Wolf, Paner and others.

That other regions far from the Nile Valley remained unexplored until recently is correct, but maybe some examples other than the case study Eastern Sudan could be mentioned?

E.g. recent work in Kordofan, where also remote sensing is essential, see Eger, Jana and Tim Karberg 2020. Nord-Kordofan im Satelitenbild: Vorbericht über die Forschungen des InterLINK-Projektes 2020. Der Antike Sudan. Mitteilungen der Sudanarchäologischen Gesellschaft zu Berlin 31, 87-98; or see the Bayuda Desert, where also geoarchaeological research was carried out, see Lohwasser, Angelika, Tim Karberg, and Johannes Auenmüller (eds), Bayuda studies: proceedings of the first international conference on the archaeology of the Bayuda Desert in Sudan, 2018.

Domed tombs in another area of Sudan (Qerri and Sabaloka region) were recently discussed, maybe this is also of interest: Siddig Babiker Ahmed Daffallah 2015. The archaeological and ethnological reconnaissance of Qerri area. In Zach, Michael H. (ed.), The Kushite world: proceedings of the 11th international conference for Meroitic studies, Vienna, 1-4 September 2008, 405-414. Vienna: Verein der Förderer der Sudanforschung.

6. PLOS authors have the option to publish the peer review history of their article (what does this mean?). If published, this will include your full peer review and any attached files.

Reviewer #1: **Yes: **Michael Kempf

Reviewer #2: **Yes: **Julia Budka

---

## [Author Response · Author response to Decision Letter 0]

18 May 2021

-The Technical Editorial Team asks to amend the Acknowledgments section and the Funding Statement in accordance with PLOS guidelines

The revised manuscript was amended accordingly.

-The Technical Editorial Team states “We note that Figures 1, 4, 7 and 9 in your submission contain map/satellite images which may be copyrighted.”, asking for a written permission by the copyright holder.

Figure 1A and 1D contain Digital Elevation Data that is freely available at German Aerospace Center (DLR) (2018): TanDEM-X - Digital Elevation Model (DEM) - Global, 90m. https://doi.org/10.15489/ju28hc7pui09.

Figure 1B is a modified extract from fully Open Access content previously published by Stefano Costanzo https://doi.org/10.1080/17445647.2020.1869112.

Figure 1C contains Landsat-8 satellite images freely available at https://www.usgs.gov/.

Figure 4 does not contain maps, it is a dataset-derived density estimation. Original content created by Filippo Brandolini with Rstudio.

Figure 7 does not contain maps, they are graphic renditions of Digital Elevation Model-derived variables employed in the analyses. Original content created by Filippo Brandolini with GRASS GIS and Rstudio.

Figure 9 does not contain maps, it is a dataset-derived fitness estimation. Original content created by Filippo Brandolini with Rstudio.

Dr. Michael Kempf suggested several language and sentence structure improvements, which we gratefully accepted - apart from very few cases discussed below. We thank him, as none of us is a native speaker!

-He states that we submitted a methodical paper, observing that during paragraph 4 (Workflow and results) <<the authors are mixing up methodical approaches and subsequent results>> and that <<The paper thus turns into a completely methodical and methodological article>>, although acknowledging the difficulty in separating workflow and results - because the workflow is a result per se when applying this kind of step-by-step geospatial analysis. He thus suggests finding a smooth way to overcome this issue.

While we understand that the article’s structure is slightly anomalous, we don’t agree on it being a completely methodological one, but rather a methodological-applicative case study whose incipit was given by the vast archaeological dataset – not the other way around. This led to new discoveries on two sides: from one hand, we investigated some aspects of the largely unknown funerary ethnoarchaeology of Eastern Sudan, while on the other hand we discovered the applicability of a previously untested statistical tool (the NSC) to humanities. With this premise, we think that our Workflow and results paragraph is a suitable depiction of the process that led us from the preliminary evaluation of our raw data to the choice of the NSC process for the modelling of the qubbas’ spatial pattern, which came upon trial and error and exhaustive theoretical preparation. During the initial writing of the article we tried to follow the classic Methods-Results-Discussion-Conclusions layout with a sharp separation between methods (steps) and results, which however spoiled the readability of the paper for any archaeologist who’s not familiar with R and advanced statistics, requiring an inevitable back and forth between paragraphs to keep track of the process.

For these reasons, we prefer not to change the structure of Paragraph 4 – Workflow and results within the manuscript. However, we tried to overcome the issue by adding a brief foreword about the benefits of Paragraph 4’s structure in the last section of Introduction (largely reworded following Rev. 2’s suggestions), stressing the concept that it is designed as a supporting narrative to the deposited R script (“…a workflow that is presented in paragraph 4 as an easily replicable step-by-step procedure”). Indications to the incremental steps of Paragraph 4 were added within the text of Paragraph 3.2 – Point Pattern Analysis in the form of upcoming paragraph numbers (lines 196, 197 and 205). 

Moreover, we would like to add that a similar approach was used by Knitter, Nakoinz, Carrero-Pazos etc. in the relevant literature that we cite in Paragraph 3.2 – Point Pattern Analysis.

GENERAL COMMENTS

ABSTRACT:

-41: behavior

Changed in the revised manuscript

-46: to decipher

Changed in the revised manuscript

-51: maybe introduce cairns as funerary elements earlier to a broader readership of PLOS One

Since we don’t use it anywhere else in the manuscript, we substituted the term “cairns” with “monuments”

-52: which is still embedded. Do you mean today?

Yes, we meant today. We reworded the sentence to prevent misunderstandings.

1 INTRODUCTION:

-57: extremely?

Removed from the revised manuscript

-60: intricate? Maybe complex or nested?

We substituted ‘intricate’ with ‘complex’ in the revised manuscript

-61: missing reference

Reference added

-62: vegetation cover?

Changed in the revised manuscript

-68: usually instead of oftentimes?

Changed in the revised manuscript, the section was reworded

-67-70: consider rewording this section

The section was reworded from 65 to 71, adding bibliographic suggestions provided by Prof. Julia Budka (Rev. 2)

-73: infrastructure

Changed in the revised manuscript

-73: subject to, prone to, vulnerable?

‘Prone to’ was used in the manuscript

-74: as much as more eminent? Like many other?

This was a subtle provocation towards the tendency of famous (to the general public) sites to overshadow lesser known ones when public awareness about conflict- and land mismanagement-driven destruction is risen. The sentence was reworded for smoother reading.

-75: One example is the hilly Kassala region and Eritrean borderland in Eastern Sudan, where…

Changed in the revised manuscript

-76: copius scatters? Maybe scattered patterns?

‘Copious scatters’ was substituted with ‘thousands’ in the manuscript

-78: that such monuments occur in forms belonging… please consider rewording this phrase

The sentence was reworded

-80: missing word. And classified?

The sentence containing this typo was reworded

-Also 80: around the shared characteristic of being round… that sounds odd. Consider rewording.

The sentence was reworded taking into consideration Dr. Kempf’s observation to line 136. The term ‘round’ was used improperly instead of circular, to describe the shape of the base of the tumuli. ‘Ring’, ‘disk’, ‘cone’, ‘heap’, as newly stated in the reworded sentence at line 136, refer to the raised portion of tumuli.

-81: are of instead of have?

Changed in the revised manuscript

-82: Another

Changed in the revised manuscript

-85: with early authors you mean: Previously, or previous authors, or previous results..

‘Initial investigations’ was used in the manuscript instead of ‘Early authors’, because the works we cite were published in 1922 and 1952 and were the first ones on the topic

-85: please reconsider the wording of ‘tribes’, which has a negative connotation. Maybe ‘groups’ is more elegant or just “Beja people”.

Noted and changed to ‘groups’ in the revised manuscript, we are grateful to the reviewer for pointing this out as it could have sparked unintended outrage.

-89: , which led to the..

Changed in the revised manuscript

-95: what is a traditional GIS approach?

’Traditional’ removed from the manuscript as it was redundant and didn’t really have a meaning within the sentence.

2 GEOGRAPHY AND ARCHAEOLOGY

-109: are located, and delete respectively

Changed in the revised manuscript

-116: erosive power? What means alternance?

’Erosive power’ added to the manuscript instead of ‘erosive action’. ‘Alternance’ was wrong and changed with ‘alternation’, meaning a sawtooth-like alternation of shallow valleys and ridges.

-117: extant? You mean the modern? I mean, you are using modern elevation data so I assume modern would be the right word.

‘Extant’ is a commonly used term in geomorphology, where it refers to palimpsests of geological/physiographic/hydrographic features resulted from past and current environmental processes that are readable in the present-day – extant – landscape.

-120: Archaeological records from the… suggest…

Changed in the revised manuscript

-122: particularly

Changed in the revised manuscript

-122: Mesolithic versus Mesolithic and Palaeolithic, be concise in the paper.

’Mesolithic’ was substituted with ‘prehistoric’ at line 122

-124: refer to the original references here!

We thank Dr. Kempf for pointing out that the original references may be of great interest for the reader, but because (Manzo, 2017) is a seminal monography that is mostly based on yearly reports and sites gazetteers from the past 40 years we prefer not to include too much minor bibliography. The most relevant references contained therein were already included in our manuscript [23-31].

-125: what do you mean with relatively sedentary? Reconsider this methodologically.. ‘relatively’ is not a very precise expression in scientific work..

Dr. Kempf’s observation is correct. ‘Relatively’ was removed from the manuscript, and the statement was slightly reworded with the addition of a supporting reference regarding prehistoric sorghum domestication in the region. 

-136: in line 80 you state that they share round characteristics! This is contradictory to what you say here…

The sentence was reworded to match with line 80. 

-140: According to previous results (citation…)

Suggestion accepted but we preferred to say ‘According to previous observations’, which seems more appropriate for the content of those publications, which were in fact brief notes on field surveys.

-141: they not there

Changed in the revised manuscript

-142: Two-storey domed

Changed in the revised manuscript

-142-144: maybe use century and not secolo or sec. maybe use 16th century AD -> check the journal guidelines for this.

Changed in the revised manuscript accordingly

-144-145: Their spatial extent stretches from… to or: They stretch from… to

See ‘144-147’ below

-145: in the north

See ‘144-147’ below

-144-147: this is grammatically wrong and hard to understand. Please rephrase.

The sentence was rephrased and simplified. In the revised version we indicated the mentioned locations with geographic coordinates, instead of distances and directions from Jebel Maman. 

-151: satellite views is certainly not correct. Imagery or images

Changed in the revised manuscript

3 MATERIAL AND METHODS

Following Dr. Kempf suggestion, we added few more references along the paragraph 3 “Material and Metods”. In particular, the fundamental work of Carrero Pazos et al. 2019 (see References).

4 WORKFLOW AND RESULTS

-216: A Kernel Density…

Changed in the revised manuscript

-216-219: please add references

Added in the revised manuscript

-242/258/9: igneous vs Igneous and metamorphic. Be concise.

We are not sure about this observation; is Dr. Kempf asking to avoid using capital and lowercase letters for the same terms is various parts of the manuscript? If this is the case, we uniformed to lowercase. 

-357 add model before hybrid

Changed in the revised manuscript

-366: site’s ? the site’s distribution? Better: site distribution patterns?

’Spatial patterning of sites’ distribution’ was changed with ‘sites’ distribution pattern’ 

-369 and others: L-function, K-function instead of L function. Please be concise

Changed in the revised manuscript

-379: add the before L-function

This sentence was removed. Please refer to ‘378-379’ for further explanation

-378-379: why do you think so? Spatial interaction only at small scale? If you just state this without adding a discussion, please cite. However, I am not totally sure that this is actually true…. I know that in quantitative methods, a radius or sigma needs to be determined, which is a totally subjective perspective but there are other many authors applying different radii in their research. This is also dependent on terrain permeability and accessibility and hence needs to be reconsidered – at least with a (small) literature review…

Limiting the radius max to 1.5 Km was merely a “strategic” choice for the plot visualization and to reduce the computing time of the process. However, we agree that this part could be misinterpreted by readers, so we decided to remove it completely and to calculate L and cross G functions without any radius limit. We updated the manuscript with the new images (revised Fig. 8 and 10), and we amended the R script code as well. The results are substantially the same, but more thoroughly represented! We are grateful to Dr. Kempf for this suggestion.

-391: its instead of their

Changed in the revised manuscript

-437: to reveal. Delete again, delete indeed, exists, resulting, weathering processes. Consider also rewording this whole sentence.. maybe split it in two?

The sentence was simplified and reworded as suggested by Dr. Kempf

-490: again: tribes. Please use groups

Changed in the revised manuscript

FIGURES:

-Fig 1: Consider changing the size and the colors of the archaeological features in the map to increase readability (although I am aware that the manuscript will be published online).

We agree with this suggestion: the Landsat base satellite image has been slightly desaturated, and the coloured symbols of the archaeological dataset made brighter. This should increase the readability without having to zoom into the picture.

-Fig. 3: please add north arrow and a grid

This Figure already has north arrow and a scale! They are a bit small, but they are clearly visible in the 300dpi .tiff picture we uploaded…maybe the reviewer is looking at the scaled-down pictures used to build the pdf for the review.

On our own initiative, we lightly revised Fig. 4 (“(ppl)” was deleted from the headings) and Fig.6 (the layout of the histograms was changed to improve readability).

Moreover, all the originally submitted images were processed through the Preflight Analysis and Conversion Engine (PACE) for optimization, as recommended by PLOS.

Prof. Julia Budka asks to elaborate on, and maybe reconsider, a few key points of our research and word choices. Moreover, she provided precious insights and literature to include in our manuscript in order to reference to other regions in Sudan, where our proposed research method may be applicable due to comparatively similar amounts of archaeological evidence or remoteness – thus needing partly desk-based research.

-Line 94 and passim: Could you elaborate why „monumental funerary landscape of Eastern Sudan“ is an appropriate term? Why and how are these tombs monumental? What is your reference point here, thinking of other monumental stone monuments etc.

When we say ‘monumental funerary landscape of Eastern Sudan’ we do not mean “a landscape containing large monuments”, but rather the impressive landscape created by the sheer amount of tumuli and qubbas, as much as the striking effect that such landscape has on the traveller when observed within the scenery of the region. While it is true that neither the tumuli nor the qubbas, taken individually, are monumental - speaking of size, ornaments etc -, when observed as a clustered unity of thousands of elements spreading across several kilometres at the foothill of the rocky mountains of the Red Sea Hills they can truly be considered a monumental manifestation.

In an effort to avoid cluttering, this explanation was achieved in the revised manuscript by rearranging and partially rewording the last section of the Introduction paragraph, with the addition of some recent literature on stone structures/burials/megaliths in the MENA regions.

-Line 100: Maybe you want to refer for the „formation of funerary landscaped in arid regions“ to recent work in the Fourth Cataract or in the Bayuda? See below

All references recommended by Prof. Budka have been included in the revised manuscript within various sections of the Introduction.

-Line 416: Since you employed the NSC models, I completely understand that it was tempting to use „galaxy-like clusters“ for your tomb settings – but could you maybe explain what „galaxy-like“ really means here? The same holds true for line 511 “cosmogony” – as much as I like this, is it really an appropriate term for the burialscape of Eastern Sudan? If so, why?

We chose this self-explanatory comparison because the NSC tool was originally designed for the field of cosmology, but Prof. Budka’s observation prompted us to add an explanation at line 416 to prevent misleading for the future reader. We believe that ‘galaxy-like clusters’ is appropriate in our case-study because, as freshly explained in the revised manuscript, hundreds of offspring tombs were found to gravitate around very few undetectable (invisible) parents, in turn brought together by the constraints of the natural settings - like stars revolve in galaxies that form higher hierarchical gravitational systems. With this new addition, and another brief recall to the concept at line 517, the expression ’Cosmogony of burials’ as part of the Conclusions Paragraph title in our opinion becomes fully legitimate without the risk of being misinterpreted as an astroarchaeology attempt.

-Line 417: Could you elaborate what the precise socio-cultural choices were? In what respect are they “precise”? See also your conclusion where a number of questions are still left open and will only be answered by fieldwork.

Prof. Budka’s question opens to interesting considerations, but honestly we used the term ‘precise’ improperly, not realizing it was an Italian-English calque… Lacking a thorough ethnoarchaeological assessment (as stated at line 494 original manuscript) we cannot speculate on the topic, and given the general meaning of the sentence where the term ‘precise’ was included we opted to just remove it to make the paragraph match with the open questions of our conclusions.

-Line 498ff: Are there any means to explain the comparably very low number of tumuli compared to qubbas? How those this difference in quantities affect or not affect your results and proposed narrative of a continuous use/occupation?

In general, the lower number of tumuli, compared to qubbas, may reside in a hypothetically lower population density at the time of their realization, but the lack of information on this topic (line 503 original manuscript) allows no speculation. Nevertheless, we think that it does not affect our discussion and conclusions because, despite their lower number, the tumuli are evenly distributed in our selected area, which is geographically unitary throughout. Moreover, we would like to add that, while remotely compiling the dataset, the possible erosion/destruction of tumuli (as well as qubbas) was of course taken into consideration. We tackled the issue by comparing multiple hi-res satellite images with field-observed badly eroded monuments, achieving the successful identification of elements comprising all grades of conservation. When we compare our computed results with the available historical (Dahl & Hjort-af-Ornas 2006) and ethnographic (Krzywinski 2012) works – all cited in the manuscript – we can reasonably consider our narrative of continuous use convincing despite the lower number of tumuli, as no explicit occupation gap appears to have taken place in the region for the past 2000 years.

---

## [Editor Report · Decision Letter 1]

7 Jun 2021

Creating the funerary landscape of Eastern Sudan

PONE-D-21-07238R1

Dear Dr. Costanzo,

We’re pleased to inform you that your manuscript has been judged scientifically suitable for publication and will be formally accepted for publication once it meets all outstanding technical requirements.

Kind regards,

Radu Iovita

Academic Editor

PLOS ONE

Additional Editor Comments (optional):

Thank you for your hard work and earnest engagement with the reviewers' comments. I have only one final suggestion, regarding the use of the cosmogony metaphor: although it is much more clearly explained in this version, I think there is still a chance that the reader will think an archaeoastronomical analysis is being introduced at the very end of the paper and be confused. While inventing a new term for 'genesis of a funerary landscape'  might make things even more confusing, perhaps something like mentioning explicitly that this is a metaphor or adding quotation marks around the word 'cosmogony' might convey your meaning sufficiently to remove any doubt. 
---

## [Editor Report · Acceptance letter]

11 Jun 2021

PONE-D-21-07238R1 

Creating the funerary landscape of Eastern Sudan 

Dear Dr. Costanzo:

I'm pleased to inform you that your manuscript has been deemed suitable for publication in PLOS ONE. Congratulations! Your manuscript is now with our production department. 

Kind regards, 

on behalf of

Dr. Radu Iovita 

Academic Editor

PLOS ONE